# Topological control of liquid-metal-dealloyed structures

Longhai Lai[1], Bernard Gaskey[2], Alyssa Chuang[2], Jonah Erlebacher[2] & Alain Karma [1✉]

The past few years have witnessed the rapid development of liquid metal dealloying to fabricate nano-/meso-scale porous and composite structures with ultra-high interfacial area for diverse materials applications. However, this method currently has two important limitations. First, it produces bicontinuous structures with high-genus topologies for a limited range of alloy compositions. Second, structures have a large ligament size due to substantial coarsening during dealloying at high temperature. Here we demonstrate computationally and experimentally that those limitations can be overcome by adding to the metallic melt an element that promotes high-genus topologies by limiting the leakage of the immiscible element during dealloying. We further interpret this finding by showing that bulk diffusive transport of the immiscible element in the liquid melt strongly influences the evolution of the solid fraction and topology of the structure during dealloying. The results shed light on fundamental differences in liquid metal and electrochemical dealloying and establish a new approach to produce liquid-metal-dealloyed structures with desired size and topologies.

[1] Physics Department and Center for Interdisciplinary Research on Complex Systems, Northeastern University, Boston, MA 02115, USA. [2] Department of Materials Science and Engineering, Johns Hopkins University, Baltimore, MD 21218, USA. ✉email: a.karma@northeastern.edu

Dealloying has grown into a powerful and versatile technique to fabricate nano-/meso-scale open porous and composite structures with ultra-high interfacial area for diverse functional and structural materials applications such as catalysts[1,2], fuel cells[3,4], electrolytic capacitors[5,6], radiation-damage resistant materials[7], high-capacity battery materials with improved mechanical stability[8,9], or composites with superior mechanical properties[10,11]. In its various forms, dealloying involves the selective dissolution of one element of an initially structureless "precursor alloy" into an external medium, causing the undissolved alloy elements to reorganize into a structure with non-trivial topology and different composition than the initial alloy. While traditional electrochemical dealloying (ECD), which uses an electrolyte as external medium, has been the most studied to date[12], this technique limits the dealloyable alloy system (e.g. Ag-Au or Ni-Pt) to contain a relatively noble element (Au, Pt) with a large enough difference in reduction potential to enable porosity formation. A major step to overcome this limitation came with the recent rediscovery of liquid metal dealloying[13,14] (LMD) that uses a liquid metal (e.g. Cu, Ni, Bi, Mg etc) as external medium, thereby enabling selective dissolution of other elements from diverse alloys (e.g. TaTi, NbTi, FeCrNi, SiMg, etc)[6,8,10,11,14–19]. Both LMD, and its variant solid metal dealloying (SMD) operating at a lower temperature where the host metal is solid[20,21], yield composites of two or more interpenetrating phases that can be transformed into open porous structures after chemical etching of one phase. Dealloying techniques have been further enriched by the even more recent introduction of vapor phase dealloying (VPD), which uses differences in vapor pressure of solid elements to form open nanoporous structures by selective evaporation of one element[22,23].

At a qualitative level, all these dealloying techniques share two important common features of the self-organizing dealloying process. The first is the aforementioned selective dissolution of an alloy element, e.g. $B$ in the simplest case of a $A_X B_{1-X}$ alloy, into the external medium. The second, first highlighted in pioneering experimental and theoretical studies of ECD[24], is the diffusion of the undissolved element A along the interface between the alloy and the external medium during dealloying. Diffusion enables formation of A-rich regions by a process similar to spinodal decomposition in bulk alloys, albeit confined at an interface. Despite those similarities, different dealloying techniques can produce different morphologies for reasons that are still not well understood[18]. While ECD can produce high-genus topologically connected structures for an atomic fraction ($X$) of the undissolved element (e.g. Au in AgAu) as low as 5%[25], computational and experimental studies of LMD have shown that this seemingly similar technique only produces topologically connected bicontinuous structures for significantly larger $X$, e.g. ~20% in the case of TaTi alloys dealloyed by a Cu melt (see Fig. 2 in Ref.[18] for a side by side comparison of ECD and LMD morphologies with varying $X$). This difference was theoretically interpreted in terms of a diffusion-coupled growth mechanism, distinct from interfacial spinodal decomposition, and closely analogous to eutectic coupled growth[26]. In a dealloying context, diffusion-coupled growth enables A-rich filaments (or lamellae in 2D) and B-rich liquid channels to grow cooperatively by diffusion during the dealloying process[15]. Coupled growth gives rise to aligned topologically disconnected structures for intermediate $X$ and is suppressed for lower $X$ where only disconnected islands of the A-rich phase can form. For larger $X$, coupled growth becomes unstable, promoting the formation of desirable connected 3D structures that maintain structural integrity even after etching of one phase. Interestingly, aligned structures produced by LMD[17] or SMD[20] of $(Fe_{80}Cr_{20})_X Ni_{1-X}$ alloys have been experimentally observed for $X$

as large as 0.5, suggesting that diffusion-coupled growth is a ubiquitous mechanism for LMD and SMD, but not ECD that typically produces porous structures with no preferred alignment.

To elucidate the origin of this difference between ECD and LMD morphologies, we carried out a combined phase-field modeling and experimental study of LMD of $Ta_X Ti_{1-X}$ alloys in which dissolution kinetics is altered by the addition of a solute element to liquid Cu. We reasoned that, even though ECD and LMD are both controlled by selective dissolution and interface diffusion, the two processes also have important differences that can potentially contribute to morphological differences[18]. First, dealloying kinetics in ECD is interface-controlled with a constant dealloying front velocity $V$ that depends on applied voltage[12]. This is true even upon addition of a small fraction of a high melting point species to the precursor alloy (e.g., Pt to Ag-Au) that slows interface mobility, refining and stabilizing the dealloyed material, but otherwise maintains the same morphology[27]. Topologically connected structures are only obtained at small $X$ for low $V$ with the large retention of the miscible element[25] to keep the solid volume fraction sufficiently large to prevent fragmentation of the structure. This suggests that dissolution rate relative to interface diffusion may play an essential role in morphology selection. In contrast, dealloying kinetics in LMD is diffusion-controlled[15,16] and comparatively faster with velocity decreasing in time as $V \sim \sqrt{D_l/t}$ where $D_l$ is the liquid-state diffusivity of the miscible element.

Second, in ECD, the immiscible element has a vanishingly small solubility in the electrolyte and hence can only diffuse along the alloy-electrolyte interface. In contrast, in LMD, the "immiscible" element (A) of the precursor $A_X B_{1-X}$ alloy typically has a small, albeit finite, solubility in the melt. This small solubility can be deduced from a ternary phase diagram analysis for the ternary CuTaTi system presented in Supplementary Fig. 1. Solubility can be quantified by plotting the liquidus line relating the equilibrium Ta and Ti concentrations on the liquid side of the interface ($c_{Ta}^l$ versus $c_{Ti}^l$, respectively) at the dealloying temperature (Supplementary Fig. 1b). Since the solid-liquid interface remains in local thermodynamic equilibrium during dealloying, $c_{Ti}^l$ is approximately constant with a value inversely related to $X$. Supplementary Fig. 1b shows that $c_{Ta}^l$ falls in the range $10^{-3} - 10^{-2}$ for typical values of $c_{Ti}^l$ measured experimentally in the CuTaTi system[15,16]. This "leak" of the immiscible element from the alloy can both influence interfacial pattern formation at the dealloying front and contribute to the dissolution and coarsening of the structure by bulk diffusion.

To assess separately the contributions of (i) reduced dealloying velocity $V$ and (ii) reduced leak rate of the immiscible element into the melt, we proceeded in two steps. First, since $V \sim \sqrt{D_l/t}$, the effect of reduced $V$ can already be checked with a pure Cu melt by investigating the morphological evolution of the structure at the dealloying front at sufficiently large time. We therefore investigate this effect by running phase-field simulations to much larger time than in a previous study that revealed the existence of topologically disconnected aligned structures formed by diffusion-coupled growth for intermediate $X$[15]. Second, to investigate the effects of reduced leak rate of the immiscible element, we add separately to Cu melts Ti and Ag that increase and reduce the leak rate, respectively, and investigate both computationally and experimentally the resulting morphologies, dealloying kinetics, and concentration profiles inside the dealloyed structures. We add Ti in amounts varying from 10% to 30% to the Cu melt dealloying medium. Ti addition raises the Ti concentration at the edge of the dealloyed layer, which reduces the Ti concentration gradient inside that layer and decreases the dissolution rate. It also increases the Ta leak rate by increasing $c_{Ti}^l$

and hence $c_{Ta}^l$ (Supplementary Fig. 1b). We add Ag in amounts varying from 10% to 30%. Since the main effect of Ag addition is to lower the melt solubilities of alloy elements, we modeled the quaternary CuAgTaTi system as an effective ternary (CuAg)TaTi system with Ti and Ta solubilities that depend on Ag concentration in the CuAg melt (See Supplementary Note 2 and Supplementary Figs. 2–4). Ag addition does not elevate the Ti concentration at the edge of the dealloyed structure. However, due to the lower solubility of Ti in Ag compared to Cu, it reduces $c_{Ta}^l$ (Supplementary Fig. 4b) and hence the Ta leak rate.

The phase-field simulation results show that coupled growth becomes unstable for large enough time, thereby promoting the formation of topologically connected structures at the dealloying front. We confirm this finding experimentally by showing that the bottom layer of a dealloyed Ta$_{15}$Ti$_{85}$ alloy, which formed close to the dealloying front at the late stage of dealloying, remains topologically connected after etching of the Cu rich phase. Our results further show that the leak rate has a profound effect on morphological evolution due to bulk diffusive transport of the immiscible element in the liquid melt. This effect, absent in ECD, is shown here to strongly influence the concentration profiles of

different elements within the dealloyed layer, the solid fraction, and the topology of LMD structures.

## Results and discussion

**Phase-field simulations.** In this section, we first present the results of the investigation by phase-field simulations of the effects of adding Ti or Ag to Cu melts, which yields dramatically different morphologies. Figure 1 shows the results of 3D phase-field simulations of Ta$_X$Ti$_{1−X}$ alloys of low atomic fraction of the immiscible element ranging from 5% to 15% dealloyed by Cu$_{70}$Ti$_{30}$, Cu$_{70}$Ag$_{30}$, and pure Cu melts. The top two rows show that both Ti and Ag addition promotes the formation of topologically connected structures compared to disconnected structures for pure Cu (third row). However, Ti addition increases as expected the Ta leak, thereby preventing dealloying for low $X$ (Ta$_5$Ti$_{95}$ and Ta$_{10}$Ti$_{90}$) and causing substantial dissolution of the dealloyed porous layer during dealloying for Ta$_{15}$Ti$_{85}$. In contrast, Ag addition (second row) promotes the formation of a topologically connected structure for all base alloy compositions with negligible dissolution of the dealloyed layer. The formation of a bicontinuous structure is further illustrated in Fig. 1b that shows

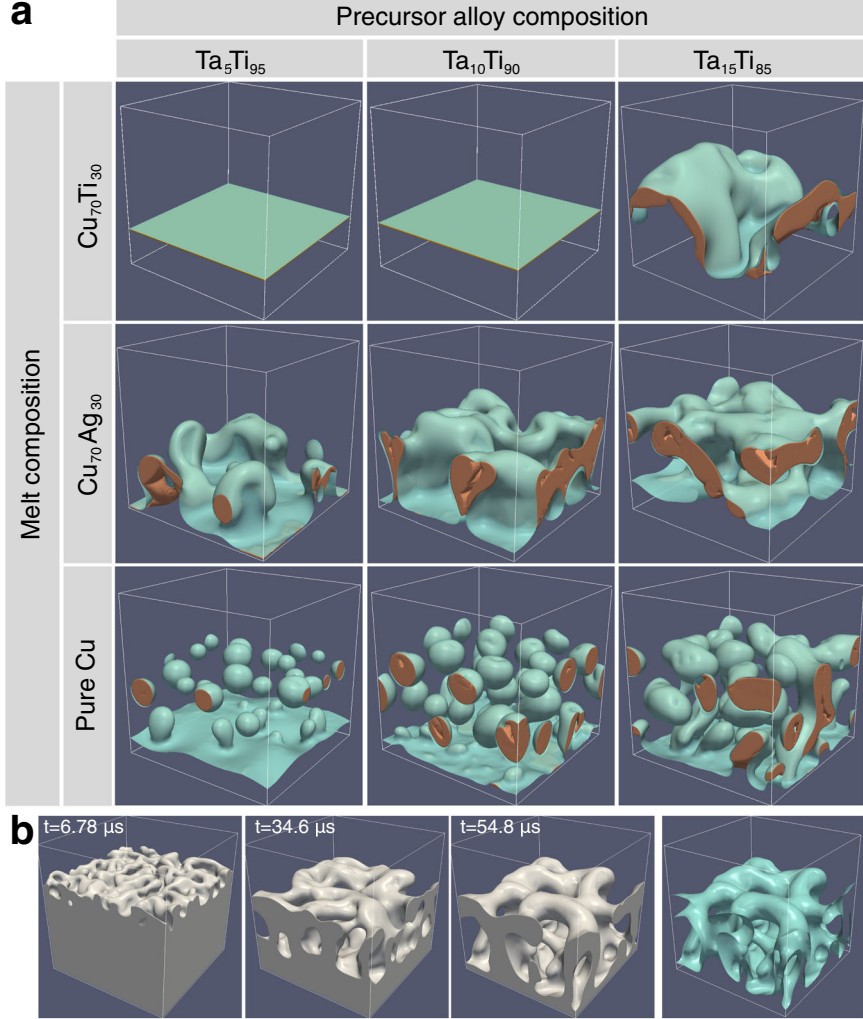

**Fig. 1 Effect of melt composition on dealloyed structure topology. a** 3D phase-field simulations (128 × 128 × 128 nm³) showing the dramatic effect of solute addition to the liquid melt on final dealloyed morphologies. The top labels denote the compositions (Ta$_X$Ti$_{1-x}$) of the precursor alloy and the vertical labels denote the melt composition of the Cu based dealloying medium. The brown color represents the high Ta concentration regions inside the dealloyed structure and the solid-liquid interface is shown in cyan color. **b** 3D phase-field simulation (190 × 190 × 190 nm³) of the precursor alloy Ta$_{15}$Ti$_{85}$ dealloyed in the Cu$_{70}$Ag$_{30}$ melt. The first 3 frames show the solid region of the dealloyed structure at different dealloying depth and the last frame shows only the solid-liquid interface at the largest depth. A movie corresponding to (**b**) is given in the Supplementary Movie 1.

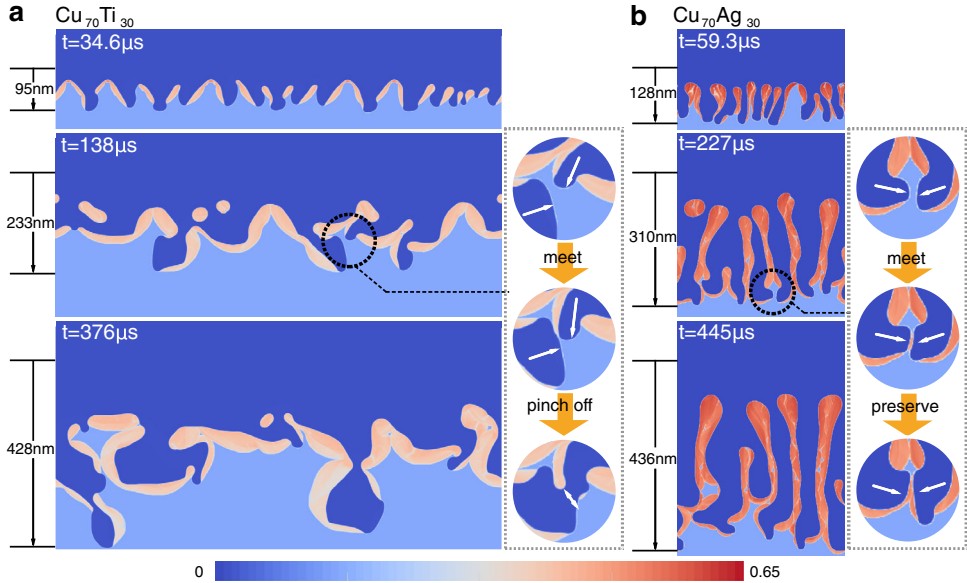

**Fig. 2 Interface dynamics controlling topology selection.** Snapshots of 2D phase-field simulations of dealloying of $Ta_{15}Ti_{85}$ alloys by $Cu_{70}Ti_{30}$ (**a**) and $Cu_{70}Ag_{30}$ (**b**) melts illustrating unstable diffusion-coupled growth. Snapshots are shown at different dealloying depths measured from the initial position of the planar solid-liquid interface. The insets illustrate different modes of collisions of liquid channels giving rise to solid ligament pinch off and preservation for $Cu_{70}Ti_{30}$ and $Cu_{70}Ag_{30}$ melts, respectively. The domain width of $Cu_{70}Ti_{30}$ is 1024 nm and $Cu_{70}Ag_{30}$ is 384 nm. The color bar refers to the Ta concentration and different colors distinguish the liquid region (dark blue), base alloy (light blue), and dealloyed structure (near red). Movies of those simulations highlighting the convoluted paths of penetrating liquid channels during unstable diffusion-coupled growth are given in the Supplementary Movies 2 and 3.

images of dealloyed structures at increasing dealloying depth increasing from left to right, and an image of the solid-liquid interface at the largest depth (right-most image).

The effects of solute addition were further investigated by 2D phase-field simulations that provide additional insights into interfacial pattern formation at the dealloying front and can access larger length and time scales than 3D simulations to extract quantitative information on dealloying kinetics. Figure 2 show snapshots of simulations of dealloying of a $Ta_{15}Ti_{85}$ precursor alloy by $Cu_{70}Ti_{30}$ and $Cu_{70}Ag_{30}$ melts. In both cases, diffusion-coupled growth is strongly unstable. Instead of penetrating vertically into the alloy, as during stable growth that favors aligned structures[15], the tips of liquid channels chaotically move sideways left and right following highly convoluted paths that, in 3D, promote the formation of topologically connected structures (Fig. 1). There is, however, one important difference between Ti and Ag addition. For the $Cu_{70}Ti_{30}$ melt (Fig. 2a), coalescence of solid-liquid interfaces from the collision of two liquid channels causes the solid ligament entrapped by the two channels to pinch off from the structure and ultimately dissolve. In contrast, for the $Cu_{70}Ag_{30}$ melt (Fig. 2b), coalescence is prevented by Ta enrichment of the solid-liquid interfacial layers due to reduced Ta leak into the melt. As a result, ligament pinch off at the dealloying front is suppressed, thereby promoting the formation of a connected structure. Interestingly, when pinch off is suppressed, the chaotic swaying motion of liquid channels produces a 2D structure (Fig. 2b) with some degree of alignment. This alignment, however, does not result from stable coupled growth. In 3D, unstable penetration produces a connected bicontinuous structure without alignment (Fig. 1b).

Additional results of 2D phase-field simulations are shown in Fig. 3. Plots of dealloying depth versus time in Fig. 3a (with slopes equal to $V$) show that addition of Ti or Ag to Cu melts slows down dealloying kinetics as expected. Figure 3b shows that this slowdown is caused by a reduction of the Ti concentration

gradient in the liquid inside the dealloyed layer. It further shows that Ti (Ag) addition increases (decreases) the Ti concentration on the liquid side of the interface ($c_{Ti}^l$), thereby causing the Ta leak, as measured by the fraction of Ta dissolved in the melt as a function of time (Fig. 3c), to be increased (decreased) by Ti (Ag) addition. Figure 3d shows that, for both solutes, the solid volume fraction remains above the threshold for forming bicontinuous topologically connected structures[28–30]. While adding Ti into the melt increases the Ta leak, it also increases the retention of Ti in the solid ligaments due to phase equilibrium, thereby increasing the volume fraction to maintain the connectivity of the dealloyed structure. Our simulations roughly match the experimental measurements of the volume fraction at the dealloying front.

As the dealloying front velocity decreases in time, the effect of reducing the dealloying rate is revealed by the morphological evolution during dealloying. In a previous phase field study, we observed eutectic-like coupled growth giving rise to aligned topologically disconnected structures during dealloying of a $Ta_{15}Ti_{85}$ precursor alloy by a pure Cu melt[15]. However, running the same phase-field simulation for much longer time reveals (see Supplementary Movie 4) that coupled growth becomes unstable when the dealloying front velocity becomes sufficiently small. Instability is manifested by a lateral swaying motion of the lamellae that suppresses their alignment thereby promoting topologically connected structures. The transition from stable coupled growth to unstable swaying growth occurs at around $x_i = 250$ nm, where the velocity is 4.7 mm/s. In contrast, the corresponding dealloying depth $x_i$ of the same velocity for the $Cu_{70}Ti_{30}$ melt is around 40 nm. Therefore, we cannot observe the similar transition in the dealloying with the $Cu_{70}Ti_{30}$ melt (See Supplementary Movie 3), since the dealloying kinetics is significantly reduced by adding 30% Ti in the melt. Finally, even though diffusion-coupled growth is unstable for the slower dealloying kinetics, the spacing $\lambda_0$ of solid ligaments at the dealloying front obeys approximately the $\lambda_0^2 V = C$ law for stable growth[15,31] where $C$ is a constant.

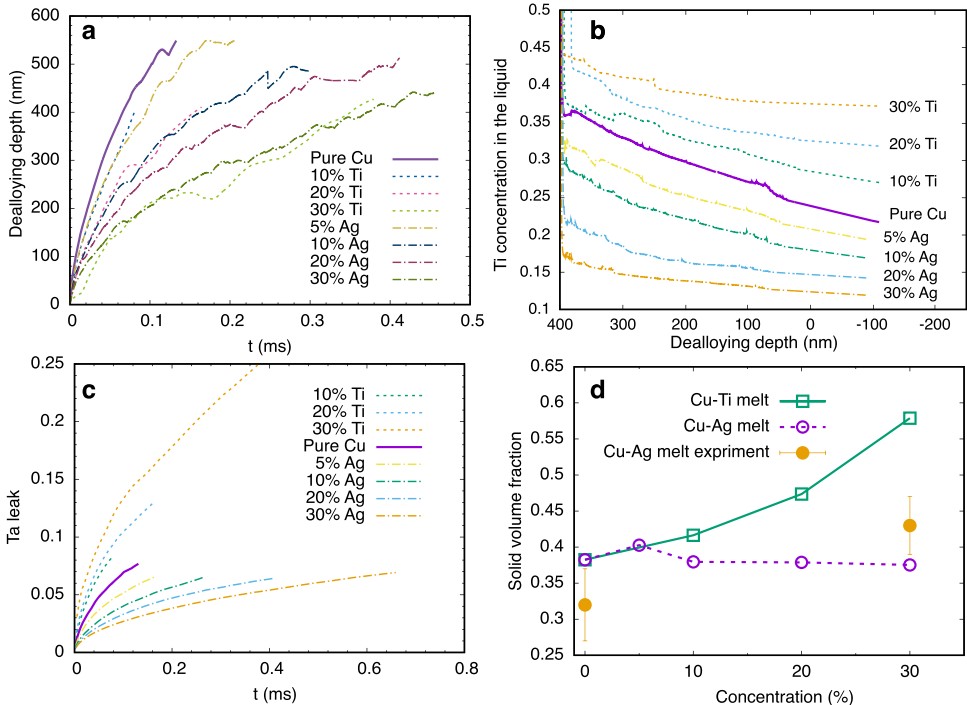

**Fig. 3 Effect of melt composition on dealloying kinetics and compositional evolution.** Phase-field simulations of dealloying of $Ta_{15}Ti_{85}$ alloys quantifying the distinct effects of Ti and Ag addition to Cu melts on dealloying kinetics measured by dealloying depth versus time (**a**), Ti concentration profiles in the liquid at a dealloying depth of 400 nm (Negative depths extend in the melt region outside the dealloyed structure. The dealloying front is at left). **b** Ta leak versus time (**c**), and solid fraction of dealloyed structures as a function of melt composition (**d**). The x axis in (**d**) is the concentration of the additional element (Ti for the green line and Ag for the purple line and experiment) in the melt.

**Dealloying experiments.** Dealloying experiments, which access larger sample scales and longer dealloying times, were carried out to test phase-field modeling predictions. Figure 4a is a schematic diagram to introduce the key parameters of the dealloyed structures. The total dealloying depth is $x_i$, which is the distance from the initial solid-liquid interface to the dealloying front. $h_L$ is the distance from the initial solid-liquid interface to the edge of the dealloyed structure before etching. A large $h_L$ indicates a strong leakage of Ta. From the SEM images of the dealloying samples, we can measure the size of the dealloyed structure before etching, $h_D$. However, as the melt is also solidified at room temperature, the dealloyed structure may be preserved without connection. Therefore, we etch out the melt (Cu rich phase) to obtain the connected structure and use $h_C$ to quantify the thickness of the connected structure.

The cross-section of the dealloyed structures shown in Fig. 4b, c support the main predicted effects of Ti and Ag addition to Cu melts on dealloying morphologies and kinetics. Figure 4b shows the bottom region of an SEM section (left) of a $Ta_{15}Ti_{85}$ alloy dealloyed to a depth $x_i \sim 270$ μm by immersion in pure Cu for 10 sec. On measurable experimental time scales, which are orders of magnitude longer than the time scale of phase-field simulations, the dealloying front velocity is much lower than the aforementioned threshold velocity of 4.7 mm/s below which stable eutectic-like coupled growth becomes unstable. As a result, the structure just above the dealloying front is expected to be fully topologically connected. Before etching, a thin layer of the base alloy is fully dissolved ($h_L = 20$ μm), which is due to the leakage of Ta (Table 1). After chemical etching of the Cu-rich phase (right), only a thin dealloyed layer is left ($h_C = 42$ μm), demonstrating that most of the dealloyed structure, which lost structural integrity during etching, was not topologically connected as predicted (Fig. 1a, the right-most image of the third row). Figure 4c shows the entire SEM section and 3D etched image of a $Ta_{15}Ti_{85}$ alloy

dealloyed to a depth of ~200 μm by immersion in a $Cu_{70}Ag_{30}$ melt for 10 sec. Since the dealloying depth is theoretically predicted to increase as $x_i(t) = \sqrt{4pD_lt}$ for diffusion-controlled kinetics (See Supplementary Note 4)[15,16], the reduction of dealloying depth from 270 μm to 220 μm with 30%Ag addition to the Cu melt corresponds to a reduction of Peclet number $p$ by a factor of 1.5. After chemical etching of the Cu/Ag-rich phase (right), the entire dealloyed structure maintains structural integrity ($h_C = 200$ μm), demonstrating that it is predominantly a topologically connected bicontinuous structure as predicted (Fig. 1, right-most image of the second row and entire bottom row). The results of all measurements of $Ta_{15}Ti_{85}$ base alloys dealloyed in multiple melts are summarized in Table 1. We also present the results of $Ta_{10}Ti_{90}$ base alloys dealloyed in multiple melts to support our conclusions. The measurements of the thickness of the Ta leak layer reveal that the structure dissolved in the $Cu_{70}Ag_{30}$ melt ($h_L = 0$ μm) is less than the pure Cu melt ($h_L = 20$ μm). In contrast, adding Ti into the melt dissolves much more the dealloyed structure ($h_L = 190$ μm). The reduction of the dissolution of the dealloyed structures between the pure Cu melt ($h_L = 250$ μm) and the $Cu_{70}Ag_{30}$ melt ($h_L = 150$ μm) is more significant in the dealloying of the $Ta_{10}Ti_{90}$ base alloy.

**Experimental characterization of dealloyed samples and theoretical interpretation.** To understand the effects of different melts, we present additional quantitative analyses of the experimental results in Fig. 5 (see also Supplementary Data 1). Figure 5a–b show the measured concentration profiles of the various elements along the dealloying direction for the experiments of dealloying in the pure Cu melt (Fig. 5a) and $Cu_{70}Ag_{30}$ melt (Fig. 5b). Concentrations of different elements are plotted as a function of the distance $d$ from the dealloying front to the edge

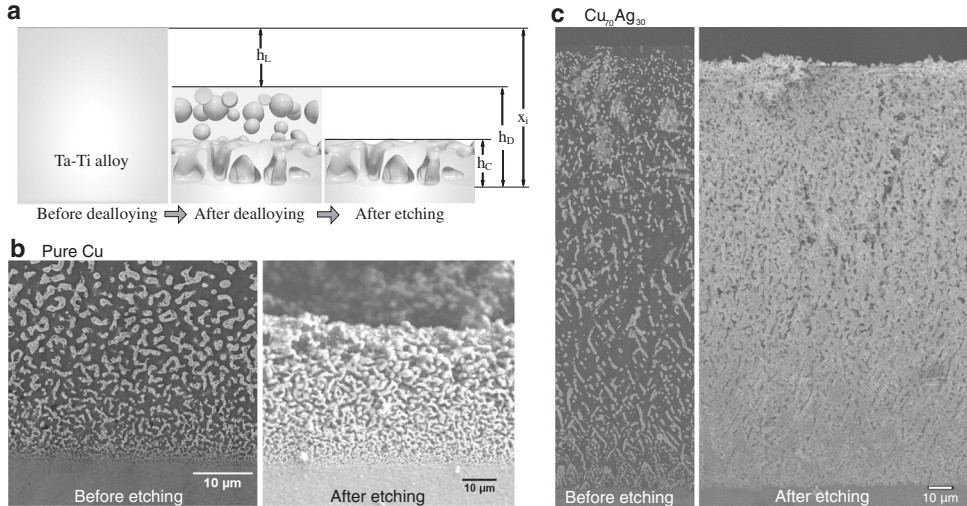

**Fig. 4 Experimental dealloyed structures. a** A schematic diagram of the morphological evolution during the dealloying process and the definition of the geometric parameters: the thickness of Ta leak $h_L$, the thickness of the dealloyed structure $h_D$, and the thickness of the connected structure $h_C$. (**b**), (**c**) Experimental validation of phase-field simulations results comparing SEM sections and 3D etched morphologies for $Ta_{15}Ti_{85}$ alloys dealloyed by pure Cu (**b**) and $Cu_{70}Ag_{30}$ melts yielding a topologically connected structure with uniform ligament size (**c**); scale bars 10 µm.

**Table 1 Measurements from dealloying experiments.**

| Base alloy | $Ta_{15}Ti_{85}$ | | | $Ta_{10}Ti_{90}$ | |
|---|---|---|---|---|---|
| Melt | pure Cu | $Cu_{80}Ti_{20}$ | $Cu_{70}Ag_{30}$ | pure Cu | $Cu_{70}Ag_{30}$ |
| $h_L$ | 20 ± 50 | 190 ± 50 | 0 ± 50 | 250 ± 50 | 150 ± 50 |
| $h_D$ | 250 ± 20 | 120 ± 20 | 220 ± 20 | 150 ± 10 | 165 ± 5 |
| $h_C$ | 42 ± 2 | - | 220 ± 20 | 75 ± 25 | 165 ± 5 |

The definition of the thickness of Ta leak $h_L$, the thickness of the dealloyed structure $h_D$, and the thickness of the connected structure $h_C$ are shown in Fig. 4a. The unit of all measurements are µm.

of the dealloyed layer in the solid ligaments and the (Cu or CuAg rich) phase that was liquid during dealloying. In contrast to ECD, where the retention of the miscible element is controlled by the dealloying rate, the concentrations in solid ligaments in LMD are determined by local thermodynamic equilibrium between the solid and liquid phases, and hence by the solid-liquid coexistence properties of the alloy phase diagram. Due to the dissolution of Ti from the base alloy, the Ti concentration decreases with increasing $d$ from the dealloying front to the edge of the dealloyed layer. As a result, the Ta concentration increases with increasing $d$ along the ligaments, which is consistent with the phase-field simulations (Supplementary Fig. 5). The decrease of Ti concentration is shallower for the $Cu_{70}Ag_{30}$ melt than the pure Cu melt consistent with slower dealloying rate. Measured concentration profiles in Fig. 5b also reveal that the ratio of Ag and Cu concentrations in the liquid is not exactly constant along the dealloyed layer, whereas in phase-field simulations this ratio is assumed constant by modeling the melt as a pseudo $Cu_{70}Ag_{30}$ element. Despite this quantitative difference, the phase-field model captures the main qualitative effect of Ag addition on the suppression of the Ta leak. Modeling fully quantitatively the concentration gradients of all four elements in the solid ligaments and liquid would require a more elaborate model of the quaternary TaTiCuAg phase diagram, which is beyond the scope of the current work.

Figure 5c compares the measured solid volume fraction $\rho(d)$ (solid lines) of the structures dealloyed by pure Cu and $Cu_{70}Ag_{30}$ melts with theoretical predictions (dashed lines) obtained from

mass conservation by using the measured Ta concentration in the solid ligaments $c_{Ta}^s(d)$ (Fig. 5a, b), and by neglecting both the Ta leak and Ta transport between ligaments at different dealloying depths. Without Ta leakage from solid to liquid, all the Ta contained in the base alloy needs to be redistributed solely to the solid ligaments. Hence, within any layer of the dealloyed structure perpendicular to the dealloying direction, mass conservation implies that $c_{Ta}^s(d)S_s(d) = c_{Ta}^0(d)S_t$, where $c_{Ta}^s(d)$ and $c_{Ta}^0$ are the Ta concentration at position $d$ in the ligaments and base alloy, respectively, and $S_s(d)$ and $S_t$ are the cross-sectional areas of the solid ligaments and the whole dealloying region, respectively. This yields the prediction for the solid volume fraction within the dealloyed layer

$$\rho(d) = \frac{S_s(d)}{S_t} = \frac{c_{Ta}^0}{c_{Ta}^s(d)}, \qquad (1)$$

which can be readily applied to the structures dealloyed by pure Cu and $Cu_{70}Ag_{30}$ melts using the corresponding $c_{Ta}^s(d)$ profiles corresponding to the blue lines in Fig. 5a and b, respectively. Those predictions, which are superimposed in Fig. 5c, show that neglecting the Ta leak gives a poor prediction of the volume fraction profiles. Mass conservation without leak predicts that the volume fraction decreases monotonously with increasing $d$, which is qualitatively observed for the pure Cu melt but not the $Cu_{70}Ag_{30}$ melt where $\rho(d)$ exhibits a minimum. In addition, it leads to a significant overestimation of the volume fraction at the dealloying front for both melts. For the smallest measurable $d \approx 10$ µm, the predicted $\rho$ values for both melts exceed 0.5 while the measured $\rho$ range from slightly more than 0.3 and 0.4 for the Cu and $Cu_{70}Ag_{30}$ melts, respectively.

To highlight the major role of the Ta leak, we next show that this quantitative discrepancy between measured and predicted $\rho$ values near the dealloying front can be resolved by refining our theoretical prediction to include this leak. For this, we compute the total number $\Delta N$ of Ta atoms that have leaked from solid to liquid when the dealloying front has moved during a time interval $\Delta t$ a distance $\Delta x_i = v \Delta t$ where $v = \dot{x}_i(t)$ is the dealloying velocity, which can be derived from the known relation $x_i(t) = \sqrt{4pD_l t}$ for the dealloying depth versus time. Local mass conservation at the dealloying front ($d \approx 0$) imposes that $\Delta N = Dg_l \Delta t S_l / v_a$, where $g_l$ is the concentration gradient of Ta atoms in the liquid, $v_a$ is the

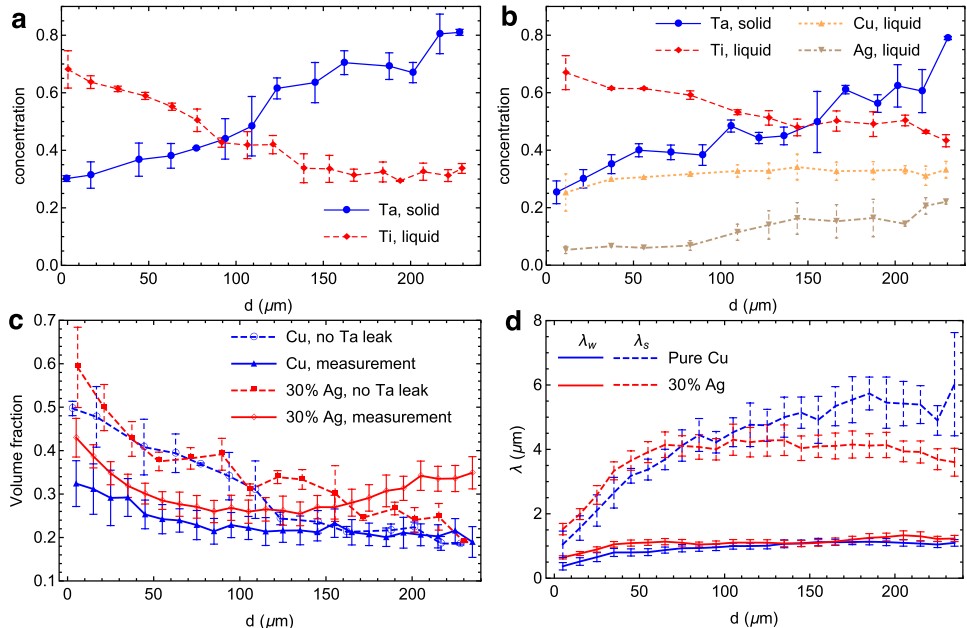

**Fig. 5 Characterization of dealloyed layers.** Measured concentration profiles as a function of distance $d$ from the dealloying front for $Ta_{15}Ti_{85}$ alloys dealloyed in pure Cu melts (**a**) and $Cu_{70}Ag_{30}$ melts (**b**). Comparison of measured solid volume fraction of dealloyed structures $\rho(d)$ (solid lines) and the theoretical prediction without Ta leak (dashed lines) corresponding to Eq. (1) (**c**); the overestimation of the prediction of Eq. (1) at the dealloying front is corrected by Eq. (2) that takes into account Ta leak. Measured average ligament width $\lambda_w$ and spacing $\lambda_s$ (**d**). Error bars represent the standard deviation.

atomic volume consistent with the definition of concentration as atomic fraction, and $S_l = S_t - S_s$ is the cross-sectional area of liquid channels at the dealloying front. The concentration gradient $g_l$ can be computed by assuming that the concentration of Ta atoms has a constant value $c_{Ta}^l$ at the interface and is vanishingly small in the melt outside the dealloyed layer, which yields $g_l = c_{Ta}^l/x_i$ and therefore $\Delta N = (\Delta x_i S_l/v_a)c_{Ta}^l/(2p)$. The solid fraction is then obtained by equating the total number of Ta atoms removed from the base alloy when the front has moved a distance $\Delta x_i$, $\Delta x_i S_t c_{Ta}^0/v_a$, to the sum of the number of Ta atoms that have leaked into the liquid, $\Delta N$, and incorporated in the solid ligament $\Delta x_i S_s c_{Ta}^s/v_a$. This equality, together with the above expression for $\Delta N$ and the relations $S_t = S_s + S_l$ and $\rho = S_s/S_t$, yields the final prediction for the solid fraction at the dealloying front

$$\rho = \frac{2pc_{Ta}^0 - c_{Ta}^l}{2pc_{Ta}^s - c_{Ta}^l}, \qquad (2)$$

which reduces to the earlier prediction without leak, $\rho = c_{Ta}^0/c_{Ta}^s$, in the limit of zero solubility of Ta atoms in the liquid ($c_{Ta}^l = 0$). Using the value of $c_{Ta}^l \approx 0.03$ from experimental measurements (not shown in Fig. 5a, b) together with the Peclet numbers $p \approx 0.26$ and $p \approx 0.17$ and solid concentrations $c_{Ta}^s \approx 0.3$ and $c_{Ta}^s \approx 0.25$ for the Cu and $Cu_{70}Ag_{30}$ melts, respectively, we obtain the predictions $\rho \approx 0.38$ and $\rho \approx 0.39$, respectively, for those two melts. Those predictions are in reasonably good quantitative agreement with measured values. The remaining differences (0.38 predicted versus 0.32 measured for the pure Cu melt and 0.39 predicted versus 0.43 measured for the $Cu_{70}Ag_{30}$ melt) can be attributed to the large uncertainty in the measurement of the very small concentration of Ta in the liquid ($c_{Ta}^l \approx 0.03$), which would be expected to be slightly larger in the pure Cu melt.

While the present experiments were carried out for specific base alloy and melt elements, we expect the insights derived from the analysis of those experiments leading to Eq. (2) to be broadly applicable to other LMD alloy systems and other related methods

such as solid-state dealloying (SSD). The effect of leak of the immiscible element on LMD structures has so far been completely ignored. This is largely because this effect is negligible in ECD, and so far LMD was naively believed to be similar to ECD. However, a crucial difference between ECD and LMD is that, in LMD, the solubility of the immiscible element in the liquid is significantly enhanced by the high concentration of the miscible element on the liquid side of the interface ($c_{Ti}^l$ here), which in turn increases the concentration of the immiscible element ($c_{Ta}^l$ here) on the liquid side of the interface and reducing the solid volume fraction as predicted by Eq. (2). This enhancement is due to the fact that the solid-liquid interface during LMD is in local thermodynamic equilibrium, such that a high $c_{Ti}^l$ contributes to raising $c_{Ta}^l$. Similarly, a high $c_{Ti}^s$ enables Cu to be incorporated in the solid ligaments and the solid Cu concentration in those ligaments decreases gradually from about 10% at the dealloying front to a negligibly small value at the edge of the dealloyed layer (Supplementary Fig. 6). In contrast, during ECD, electrochemical removal of Ag from a AgAu alloy is a non-equilibrium reaction that does not increase the solubility of Au in the electrolyte. Beyond LMD, we also expect our results to apply to SSD where the solid-solid interface is expected to remain in local thermodynamic equilibrium during dealloying. This expectation is supported by the fact that the solid volume fraction is observed to vary across the dealloyed layer of SSD structures[20], which implies that dissolution of the solid ligaments associated with leakage of the immiscible element is occurring during dealloying.

While Eq. (2) predicts a significant reduction of the solid fraction at the dealloying front due to the Ta leak, Ta transport within the dealloyed region also needs to be considered to understand the solid fraction profiles within the entire dealloyed layer, which differs markedly for the pure Cu and $Cu_{70}Ag_{30}$ melts. For the $Cu_{70}Ag_{30}$ melt (red line in Fig. 5c), $\rho(d)$ exhibits a minimum about half way through the dealloyed layer. This minimum is due to the fact that the total amount of Ta contained in solid ligaments near the edge of the dealloyed layer is larger

than in the base alloy. Namely, for $d \approx 230\ \mu m$, $S_s(d)c_{Ta}^s(d) > S_t c_{Ta}^0$, or completely equivalently, the measured $\rho(d) = S_s(d)/S_t \approx 0.35$ is much larger than the prediction of Eq. (1) without leak $c_{Ta}^0/c_{Ta}^s(d) \approx 0.2$. This implies that some of the Ta leaked is transported by diffusion in the liquid and along the solid-liquid interface, from the dealloying front to regions away from this front where it is redeposited.

This redeposition has the opposite effect of the Ta leak to enrich solid ligaments in Ta, and solid fraction profiles can be qualitatively interpreted as the balance of Ta leak and redeposition. For the $Cu_{70}Ag_{30}$ melt, the increase in Ag concentration in liquid with increasing $d$ (brown dashed line in Fig. 5b) reduces the Ta leak by decreasing the Ta solubility, thereby causing $\rho(d)$ to increase with $d$ after reaching a minimum. This maintains the solid fraction large enough to prevent fragmentation by pinching off of solid ligaments, thereby explaining why structures dealloyed in $Cu_{70}Ag_{30}$ melts maintain structural integrity after etching. In contrast, for the pure Cu melt, leak and redeposition almost balance each other giving rise to a slowly decreasing solid fraction that falls below the threshold for fragmentation over most of the dealloyed layer, leaving only a very thin layer that maintains structural integrity near the dealloying front (Fig. 4b, Table 1).

**Coarsening**. Our analysis so far has focused on explaining the strong effect of the leak of the miscible element in the dealloying medium on the solid fraction and hence the topology of dealloyed structures. We now turn to the effect of this leak on coarsening of the bicontinuous structure within the dealloyed layer, which generically occurs during LMD due to the high processing temperature. This is in contrast to ECD, where coarsening is essentially absent during dealloying but can be induced by annealing at higher temperature after dealloying. To date, coarsening during LMD has been modeled by assuming that it occurs by diffusion of the immiscible element along the solid-liquid interface, analogous to surface-diffusion-mediated coarsening of annealed ECD nanoporous structures. Accordingly, the ligament size has been modeled using the standard scaling law for capillary-driven coarsening

$$\lambda^n = k t_c + \lambda_{00}^n, \tag{3}$$

where $t_c$ is the coarsening time, defined as the time elapsed after passage of the dealloying front at depth $x_i$ within the dealloyed layer (where $\lambda$ has initial value $\lambda_{00}$) until the end of the dealloying experiment, and the scaling exponent $n = 4$ for surface diffusion. Care must be taken to use Eq. (3) to interpret measurements of $\lambda$ versus distance $d$ from the dealloying front performed on the final dealloyed structure at the end of the experiment. This is because regions closer to the edge of the dealloyed layer have had a longer time to coarsen than regions close to the front. This can be done by supplementing Eq. (3) with a relation between $t_c$ and $d$. This relation can be readily obtained using the prediction for the dealloying depth versus time, $x_i(t) = \sqrt{4pD_l t}$, which yields $t_c(d) = t_e - t_f(d)$, where $t_e$ is the duration of the entire experiment and $t_f(d) = (\sqrt{4pD_l t_e} - d)^2/(4pD_l)$ is the time at which the dealloying front reached a depth equal to the final dealloying depth minus $d$. This expression for $t_c(d)$ substituted into Eq. (3) predicts $\lambda(d)$ (see Supplementary Note 5).

To test this prediction, we performed measurements of ligament width and spacing on full cross-sections of dealloyed structures, which are exemplified for pure Cu and $Cu_{70}Ag_{30}$ melts in Supplementary Fig. 9. From line scans perpendicular to the dealloying direction performed at different distances $d$ from the dealloying front, we obtained the average width $\lambda_w(d)$ of Ta-rich ligaments and the average spacing $\lambda_s(d)$ between ligaments. Those measurements are reported in Fig. 5d and compared to the

prediction of Eq. (3) in Supplementary Fig. 10 for different values of $n$. The comparison shows that the surface diffusion exponent $n = 4$ gives a poor prediction. This prediction is not significantly improved by choosing $n = 3$ for capillary-driven coarsening mediated by bulk diffusion, which could be naively expected to provide a better fit due to Ta leak into the liquid.

This quantitative disagreement between theory and experiment is not surprising since Eq. (3) describes capillary-driven coarsening at constant volume fraction $\rho$, while, during LMD the solid fraction $\rho$ is not constant. $\rho$ varies spatially within the dealloyed layer at the end of dealloying as already shown in Fig. 5c. $\rho$ also varies in time during dealloying at fixed dealloying depth, from the value at the dealloying front, which is approximately constant in time and hence independent of $t_f$ and $d$, to the measured value of $\rho(d)$ reported in Fig. 5c corresponding to the final time $t_e$. The value at the dealloying front can be estimated from Fig. 3d to be approximately 0.4 and 0.35 for AgCu and pure Cu melts, respectively, which is higher in all cases than the final value of $\rho$ at time $t_e$. Importantly, the decrease of $\rho$ in time at fixed $d$ is a direct consequence of the existence of a concentration gradient of the miscible element (Ti) in the liquid. Since the concentration of Ti in the liquid decreases with increasing $d$, the equilibrium concentration of Ti in the solid is also a decreasing function of $d$ resulting in the dissolution of Ti from the solid ligaments and a concomitant decrease of solid fraction with time. The time variation of $\rho$ is further influenced by Ta leak and redeposition. Therefore, we expect generally coarsening during LMD to occur at non-constant volume fraction due to the additional effects of dissolution and redeposition, which can cause the structure to evolve in addition to capillary driven coarsening, and to be controlled by diffusion in the liquid and not solely along the solid-liquid interface.

The fact that Eq. (3) does not describe quantitatively the measurements of ligament width and spacing (Supplementary Fig. 10) for $3 \le n \le 4$ suggests that dissolution and redeposition, which are not driven by a reduction of interface area, plays a dominant role in the present experiments. Both $\lambda_w$ and $\lambda_s$ would be expected to have the same $d$-dependence for capillary driven coarsening, while Fig. 5d shows that $\lambda_s$ increases much more rapidly with $d$ than $\lambda_w$ for both pure Cu and $Cu_{70}Ag_{30}$ melts. While a theory of coarsening that takes into account dissolution and redeposition, which is beyond the scope of this work, would be needed to quantitatively interpret those measurements, this difference is to be qualitatively expected since complete dissolution of small ligaments promotes an increase of ligament spacing. Furthermore, the fact that $\lambda_s$ decreases towards the edge of the dealloyed layer for the $Cu_{70}Ag_{30}$ melt after reaching a maximum, but remains monotonously increasing for the pure Cu melt, can be attributed to the increase of Ag concentration in the liquid with $d$ already invoked to explain the non-monotonous behavior of $\rho(d)$ in Fig. 5c. This increase in Ag concentration with $d$ suppresses Ta leak and dissolution of ligaments, thereby causing $\lambda_s$ to decrease after reaching a maximum.

Finally, we note that computational studies of capillary-driven coarsening at constant volume fraction have shown that structures fragment during coarsening when the volume fraction is lower than a threshold of approximately 0.3[29,30]. This threshold could be slightly lower in practice since fragmentation, and the concomitant decrease of the genus, occurs on a time scale that could be comparable to or longer than the total dealloying time as in the present experiments. The fact that structures dealloyed in $Cu_{70}Ag_{30}$ melts retain their structural integrity even though $\rho(d)$ falls slightly below 0.3 over an intermediate range of $d$ suggests that fragmentation, if occurring, was only partial. It is also possible that the volume fraction threshold for fragmentation is affected by dissolution and redeposition.

Two major conclusions emerge from the present study. The first, more practical, is that the topology of dealloyed structures produced by LMD can be controlled by the choice of the melt. By choosing a melt that reduces the small, albeit finite, solubility of the immiscible element A of an $A_X B_{1−X}$ base alloy in the melt, high-genus dealloyed structures can be produced that maintain their connectivity and structural integrity even for a low concentration $X$ of this element. This was previously known to be possible for ECD[25] but not for LMD. The second conclusion, more fundamental in nature, is the reason why structural integrity can be preserved in LMD by modifying the dealloying medium, which is interesting in and of itself to explain our observations in TaTi alloys dealloyed by pure Cu and CuAg melts, but also sheds new light more broadly on important differences between ECD and LMD that were not appreciated before.

In ECD, connectedness of the structure is preserved at small $X$ by keeping the dealloying rate, which is constant in time at fixed driving force, sufficiently small to retain enough of the miscible element B in solid ligaments during dealloying to keep the solid volume fraction $\rho$ large enough to prevent fragmentation[25]. In LMD, the dealloying rate $dx_i(t)/dt = \sqrt{pD_l/t}$ decreases in time due to diffusion-limited kinetics. Hence, independently of the type of melt composition that only affects the Peclet number $p$, the dealloying rate quickly reaches a value small enough to retain enough B in solid ligaments, directly reflected in the fact that $\rho$ at the dealloying front remains approximately constant in time and above the threshold for fragmentation. As demonstrated in phase-field simulations, the dealloying rate also quickly reaches a value small enough for eutectic-like coupled growth to become destabilized, promoting the formation of topologically connected structures by a lateral swaying motion of lamellae. Therefore, the main fundamental difference between ECD and LMD lies in the evolution of the structure and $\rho$ inside the dealloyed layer after passage of the dealloying front, and not the dealloying rate.

In ECD, $\rho$ and connectivity remain constant throughout the dealloyed layer. In contrast, in LMD, both vary inside this layer as clearly revealed in the present study, which mapped atomic concentration and $\rho$ profiles throughout the entire depth of dealloyed structures produced by LMD. There are two reasons for this variation. The first is that, even in the limit of zero solubility of A, the concentration gradient of B in the liquid, absent in ECD, induces a concentration gradient of A in the solid ligaments that are in chemical equilibrium with the liquid. The gradient of A in turn induces a gradient of $\rho$ within the dealloyed layer. The second is that the leak of A into the liquid due to a non-vanishing solubility further modulates the spatial variation of $\rho$ within this layer, with decreased solubility helping to keep $\rho$ higher and more spatially uniform so as to maintain connectivity.

Finally, the evolution of ligament size and connectivity within the dealloyed layer during LMD is far more complex than surface-diffusion-limited capillary-driven coarsening at constant volume fraction, as previously thought by analogy with coarsening of annealed nanoporous ECD structures. As revealed here, coarsening in LMD occurs with a spatiotemporally varying solid fraction and is generally influenced by liquid-state diffusive transport of both A and B from the dealloying front to the edge of the dealloyed layer. The failure of scaling laws for surface- or bulk-diffusion-limited capillary-driven coarsening to describe quantitatively the variations of ligament width and spacing within the dealloyed layer suggests that transport of A and B linked to concentration gradients in the liquid plays an equally or more important role than reduction of interfacial area. The development of a theory that takes into account those various effects is an important future prospect.

## Methods

**Experiments**. Titanium-tantalum binary alloys were produced by induction melting pure Ti evaporation pellets (Kurt J. Lesker, 99.995%) and Ta evaporation pellets (Kurt J. Lesker, 99.9%) with a 45 kW Ambrell Ekoheat ES induction power supply and a water-cooled copper crucible purchased from Arcast, Inc (Oxford, ME). After melting several times, each alloy was annealed for 8 h at a temperature within 200 °C of the melting point to allow homogenization and grain growth. Samples cut from this parent ingot were attached to a Ta wire by spot welding and suspended from a manipulator arm. The metal bath was prepared by heating a 40 g mixture of Cu (McMaster Carr, 99.99%) with Ag (Kurt J. Lesker, 99.95%) or Ti pellets at high power using a 4 kW Ameritherm Easyheat induction heating system until the bath was fully molten. The power was decreased and the bath was allowed to mix and equilibrate at the reaction temperature of 1240 °C for half an hour. The manipulator arm was then lowered to immerse the sample in the bath for the specified time, before removing the sample to allow it to cool. All heating for both alloy preparation and LMD was performed in high-purity (99.999%) Ar atmosphere. After dealloying, sample cross sections were polished and observed by optical microscopy and scanning electron microscopy (SEM, JEOL JSM-6700F). Elemental analysis was performed by energy dispersive x-ray spectroscopy (EDS) in the SEM. The 3D microstructure of the dealloyed samples was observed by dissolving the solidified Cu-rich phase in 35% nitric acid solution (ACS reagent grade, Fluka).

**Phase-field model**. Simulations were carried out using a previously developed phase-field model of dealloying for ternary alloys[15]. This model couples the evolution of a phase field $\phi$, which distinguishes between the solid and liquid phases, to the concentration fields $c_i$ of alloy elements. The total free-energy of the system is expressed as

$$F = \int_V dV \left[ \frac{\sigma_\phi}{2} |\nabla \phi|^2 + f(\phi) + \sum_i \frac{\sigma_i}{2} |\nabla c_i|^2 + f_c(\phi, c_1, c_2, c_3) \right], \quad (4)$$

where $f(\phi)$ is a double-obstacle potential with minima at $\phi = 1$ and $\phi = 0$ corresponding to the solid and liquid, respectively, and $f_c(\phi, c_1, c_2, c_3)$ is the chemical contribution to the bulk free-energy density that describes the alloy thermodynamic properties. For simulating dealloying of TaTi alloys by Cu or CuTi melts, we use the same form of $f_c(\phi, c_1, c_2, c_3)$ and parameters as in Ref. [15]. For dealloying of TaTi alloys by CuAg melts, we reduce the quaternary (CuAg)TaTi system to an effective ternary system with different parameters dependent on Ag concentration as described in Supplementary Note 2. The evolution equations for the phase and concentration fields are derived in variational forms as

$$\partial_t \phi = -L_\phi \frac{\delta F}{\delta \phi}, \quad (5)$$

$$\partial_t c_i = \nabla \cdot M_{ij} \nabla \frac{\delta F}{\delta c_j}, \quad (6)$$

where $M_{ij} = M_l(1 - \phi)c_i\left(\delta_{ij} - c_j\right)$ is the atomic mobility matrix and $L_\phi$ controls the attachment kinetics of atoms at the solid-liquid interface.

## Data availability

The experimental data that support the findings of this study are available in the Supplementary Data file. The simulation parameters are available in the Supplementary Information. All data are also available from the corresponding author upon request.

## Code availability

Code for phase-field simulations is available from the corresponding author upon request.

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

## Acknowledgements

L.L. and A.K. acknowledge the support of Grant No. DE-FG02-07ER46400 from the U.S. Department of Energy, Office of Basic Energy Sciences. B.G., A.C., and J.E. acknowledge support from the National Science Foundation under grant DMR-1806342.

## Author contributions

A.K., B.G., and J.E. conceived the investigation. L.L. developed the phase-field models and performed the simulations. B.G., A.C. and J.E. conceived and designed the experiments. B.G. and A.C. prepared the samples and conducted the experiments. L.L. and A.K. wrote the paper. All the authors discussed the results and commented on the manuscript.

## Competing interests

The authors declare no competing interests.
