## [Peer Review File · Nature Communications]

Reviewers' comments:

Reviewer #1 (Remarks to the Author):

This manuscript studies liquid metal dealloying. Lai et al. attempted to demonstrate computationally and experimentally that by adding additional elements to the metallic melt, they can promote high-genus topologies and limit coarsening, for a wider range of alloy compositions. In simulation they modeled adding 1) Ti (10-30%), and 2) Ag (10-30%) to Cu melt and discussed their respective effects and mechanisms. Some experimental data were presented to compare with the simulation.

While dealloying is an important field and the topic being studied is of great scientific interest, the results remain pre-mature and conclusion unsupported in the manuscript's present form. Here are several key issues for the authors to consider:

1. On the discussion of the morphological evolution mechanisms:

1) The authors stated that "The results shed light on fundamental differences in liquid metal and electrochemical dealloying" which remains unclear to the reviewer. The authors focused on that electrochemical dealloying can produce "high genus topologies" (referring as 'bicontinuous structure' hereafter) at low atomic fraction of the undissolved element/component, while liquid metal dealloying so far showed that it can only produce bicontinuous structures at a higher atomic fraction of the undissolved component. Such difference may not reflect the fundamental differences between these two dealloying methods but rather the rate at which the dealloying proceeds?

Notably for electrochemical dealloying of low atomic fraction of the undissolved element, such as Ag-5Au, highly connected structure was formed when the materials were dealloying at a slower dealloying rate and a relatively large amount of the Ag (dissolving component) remains in the structure; only with such composition a connected network could form. E.g. H.W. Pickering in dealloying Cu-10 at.% Au showed that in addition to Cu retention, there was also a compositional gradient. Qi and Weissmueller have demonstrated taking this character to conduct a two-step dealloying with the high-retention of the Ag in dealloying Ag-5%Au. (ACS Nano, 2013). These key aspects were not clear in the discussions; related articles on dealloying Ag-5%Au should also be cited.

As adding the Ti and Ag would both result in a slower rate of the liquid metal dealloying, one key question the authors should address is that – how much Ti is retained in the bicontinuous structures dealloyed from the alloy with a lower atomic ration of Ta? This would provide a fair comparison with the electrochemical dealloying.

2) The discussion concerning the effects of 'Ta-leak' on the morphological evolution is rather confusing. Adding Ti would lead to a reduction of under-saturation of Ti in the Cu melt, and adding Ag could decrease the temperature of liquid metal dealloying and the degree of under-saturation; since both would alter the dealloying rate, it is not clear to the reviewer the exact effects of them on the morphology/topology of the dealloyed structure. If one of the main goals is to understand the attribution of 'Ta leakage' to the morphological evolution during dealloying, perhaps it would be better to simply saturate the Cu melt with Ta? This would prevent the Ta leakage and avoid affecting the rate of liquid metal dealloying of Ti, thereby more clearly test the parameter.

2. On the modeling of phase diagram and phase field simulations:

1) The information provided by the phase diagrams is rather limited and incomplete in explaining the effects of Ti and Ag additions to the Cu melt on the morphology/topology.

2) The calculation method for the phase diagram appears to be incorrect? First, as shown in equation 10, the concentration terms are missing in the calculation of the grand potential. In addition, for equations 8 & 9, the ternary phase diagrams should be calculated by equating the chemical potential of all three components, not just two?

3) In related to the phase field simulation method: As the authors noted in their phase diagram modeling, there is one independent variable left. How were they able to draw the tie lines as

shown in the phase diagrams? While not much details are given regarding the phase-field simulation, it is unclear how the authors handle this also in the phase field simulation.

4) The relationship between the diffusivity and mobility given right above the equation 7 appears to be incorrect as the interdiffusion coefficient is related to the mobility. The authors can refer to: Andersson, Jan-Olof et al, Models for numerical treatment of multicomponent diffusion in simple phases, Applied Physics (1992). As shown in eq. 43 in the reference, interdiffusion coefficients would depend on the derivative of the chemical potential with respect to concentration. While this may not affect the equations solved, it does affect how the mobilities are chosen and the interpretation of the results; the thermodynamic terms can also lead to significant off-diagonal terms in the diffusivity matrix with interesting implications for the systems discussed in the paper.

3. The comparison between the experimental and computational results are rather incomplete with some inconsistency.

1) First, experimental results did not show different alloy compositions and thus it is difficult to compare them with the computational results; the validation thus falls short.

2) Secondly, there is a lack of quantitative analysis on the experimental data. E.g. the authors stated that the additional elements in the Cu melt results in a 'smaller and spatially more uniform ligament size', is not clear to the reviewer. Such claim can hardly be observed from the SEM micrographs presented in the manuscript.

3) Elemental distribution shall be analyzed (e.g. by EDS) and compared with the simulation results, especially on the retention of the Ta per discussion in the earlier comments.

4. On the literature associated with the dealloying field:

1) First of all, the abstract stated that 'the past few years have witnessed widespread use of liquid metal dealloying' may be an overstatement. While being an important, emerging dealloying method, searching on 'liquid metal dealloying' in Web of Science as topic yielded ~ 20 papers. Out of them about half is from Erlebacher and co-authors and the other half is from Kato and co-authors, with only few from other groups. There is much to be developed in the field and stating this method as 'widespread' would be misleading.

2) Secondly, the manuscript misleadingly stated that 'the recent advent of liquid metal dealloying [13]...'; while it may be fair to state that liquid metal dealloying method has only been further explored recently to fabricate nanoporous materials as cited ref. 13 in the manuscript, it is important to note that this concept was studied in a pioneer work by Harrison and Wagner back in 1959. As such, LMD method would not be a 'recent advent' as stated. Appropriately stating the original credit of this work would be important.

Ref. Harrison, J. D.; Wagner, C. The Attack of Solid Alloys by Liquid Metals and Salt Melts. Acta Metall. 1959, 7, 722-735.

3) The authors stated that electrochemical dealloying is 'limited to relatively few alloys (e.g. Ag-Au or Ni-Pt)' does not seem to correctly reflect the work in the field. E.g. in the article published by the co-authors of this manuscript in MRS Bull in 2018 (Ref. 17), figure 1 shows that electrochemical dealloying method has been applied to more systems than the liquid metal dealloying, certainly more than the few alloys cited in this work. The authors are encouraged to present a more accurate account of the status of the field.

4) Another minor issue is that ref. 10 is not correct. It should be by McCue, Ian; Ryan, Stephen; Hemker, Kevin; et al. Adv Eng. Mat published in 2016, not 2015; the full citation is also missing.

Reviewer #2 (Remarks to the Author):

The manuscript "Topological Control of Liquid-Metal-Dealloyed Structures" describes an interesting extension of a phase field simulation method to describe and analyse the process of liquid metal dealloying. The new aspect is the addition of a second element to the liquid metal (here by adding Ag or Ti to the Cu liquid). The authors can show in their simulation that this

alloying of the corrosion agent significantly changes the diffusion kinetics at the liquid-solid interface and the solubility of the solid phase elements (Ta and Ti) in the melt what results in different dealloying front velocities and structure connectivities. Additionally, SEM pictures of similarly dealloyed structures are shown which corroborate their main findings about the influence of the alloying of the melt.

The paper is kept quite short and compact focussing on the main finding about the influence of the composition of the melt on the process. In preceding publications of the submitting authors, especially [14] and [17], the simulation technique was developed and the structure formation discussed. Under these circumstances this extreme focus might be justified and the manuscript considered for publication. But in some instances the writing might be a bit too compact making it difficult for the reader to follow. Especially the section "Results" started with detailed compositions without explaining to the reader what exactly is done. Are they writing about a simulation or a lab experiment? What is the strategy? It becomes clear, step-by-step, in the successive paragraphs but the transition should be written in a straightforward and instantly clear way.

On page 4: "suggesting that diffusion-coupled growth is a ubiquitous mechanism for ECD and SMD, but not ECD that typically produces porous structures with no preferred alignment."
...ECD and SMD, but not ECD... The first ECD probably means LMD.

"the immiscible element has a vanishingly small solubility in the electrolyte and hence can only diffuse along the alloy-electrolyte interface"

I agree with the last statement. But talking about solubility of the immiscible, i.e. remaining, element during electrochemical corrosion can be misleading. In most cases, the solubility can be controlled by the applied potential. Maybe the fact that if a part of the immiscible material goes into solution, it's not going to re-deposit on the surface (unless the potential is changed) is the more decisive point.

On page 7: "the spacing λ of solid ligaments at the dealloying front obeys approximately the $\lambda^2 V = C$ law for stable growth [14]"
This finding is only cited in reference [14]. The authors should cite the original studies.

On page 8: "corresponds to a reduction of Peclet number P by a factor of 3.3 compared to about 5.4 in the phase-field simulations."
The description of the character of the transport phenomena (flow and diffusion) by means of the Peclet number at that point is completely unmotivated. The authors should explain the reader what exactly the Peclet number describes in that scenario and what the observed change means.

On page 10: "purified Ar atmosphere": Just a technical note: Purified by which means?

Reviewer #3 (Remarks to the Author):

In this paper, influence of adding element to a metallic melt used for the dealloying medium on porous structure evolution during the liquid metal dealloying (LMD) is discussed both computationally and experimentally to find methods to reduce the parting limit concentration (X) of the remaining element for constructing the bi-continuous structure and to suppress the ligament growth.

The author group has reported that dipping a Ti-Ta alloy precursor in a Cu melt results in the selective dissolution of Ti element, thus in the self-organizing bi-continuous porous Ta in a Cu

melt. In this dealloying system, Ti and Ag were selected and added in the Cu melt separately. The phase field model simulation demonstrated that (1) the Ti addition suppressed Ti dissolution but enhanced the Ta leakage to the melt, resulting in decreasing the dealloying rate, increasing the parting limit X, and suppressing the ligament alignment by the pinch off. On the other hand, (2) the Ag addition in the melt suppressed the Ta leakage, resulting in dramatically decreasing X, and enhancing the ligament alignment by suppressing the pinch off. The different effects of these additions are successfully confirmed in the experimental counterparts. Although the paper is well organized with the data by the reliable computational and experimental protocols, and the finding in this study actually suggests methods to control the porous structure in LMD, the method itself is not special and within the range of expectations, i.e. varying dealloying temperature or time, and adding element(s) to precursor, etc., so, does not have enough impact for publication in Nature Communications.

P4, line8: “ECD and SMD, but.....” should be “LMD and SMD, but.....”

P8, line19: It is mentioned that the Ta addition to the Cu melt reducing the ligament growth rate by the bulk diffusion of Ta in the Cu liquid. It is better to mention why the bulk diffusion of Ta in the Cu melt influences the ligament growth.

Reviewer #1 (Remarks to the Author):

This manuscript studies liquid metal dealloying. Lai et al. attempted to demonstrate computationally and experimentally that by adding additional elements to the metallic melt, they can promote high-genus topologies and limit coarsening, for a wider range of alloy compositions. In simulation they modeled adding 1) Ti (10-30%), and 2) Ag (10-30%) to Cu melt and discussed their respective effects and mechanisms. Some experimental data were presented to compare with the simulation. While dealloying is an important field and the topic being studied is of great scientific interest, the results remain pre-mature and conclusion unsupported in the manuscript's present form. Here are several key issues for the authors to consider:

1. On the discussion of the morphological evolution mechanisms:

1) The authors stated that "The results shed light on fundamental differences in liquid metal and electrochemical dealloying" which remains unclear to the reviewer. The authors focused on that electrochemical dealloying can produce "high genus topologies" (referring as 'bicontinuous structure' hereafter) at low atomic fraction of the undissolved element/component, while liquid metal dealloying so far showed that it can only produce bicontinuous structures at a higher atomic fraction of the undissolved component. Such difference may not reflect the fundamental differences between these two dealloying methods but rather the rate at which the dealloying proceeds? Notably for electrochemical dealloying of low atomic fraction of the undissolved element, such as Ag-5Au, highly connected structure was formed when the materials were dealloying at a slower dealloying rate and a relatively large amount of the Ag (dissolving component) remains in the structure; only with such composition a connected network could form. E.g. H.W. Pickering in dealloying Cu-10 at.% Au showed that in addition to Cu retention, there was also a compositional gradient. Qi and Weissmueller have demonstrated taking this character to conduct a two-step dealloying with the high-retention of the Ag in dealloying Ag-5%Au. (ACS Nano, 2013). These key aspects were not clear in the discussions; related articles on dealloying Ag-5%Au should also be cited.

As adding the Ti and Ag would both result in a slower rate of the liquid metal dealloying, one key question the authors should address is that – how much Ti is retained in the bicontinuous structures dealloyed from the alloy with a lower atomic fraction of Ta? This would provide a fair comparison with the electrochemical dealloying.

This is a more subtle question because the velocity varies during the LMD process but not the ECD process. The velocity always becomes small after a long time thereby destabilizing the dealloying front and promoting a topologically connected structure. We understand that the reviewer's example demonstrates that slower kinetics promotes the formation of topologically connected structures in ECD. The mechanism is that slower kinetics promotes high retention of the miscible element, which increases the solid volume fraction thereby preserving the integrity of the bicontinuous structure.

However, in LMD, we cannot obtain a connected structure from the base alloy with a lower immiscible element fraction even after a long time where the dealloying velocity is much lower. The slower kinetics is not adequate to explain the difference between ECD and LMD. The retention of the miscible element away from the dealloying front is mainly controlled by the local chemical equilibrium (phase diagram) with the liquid phase.

The more fundamental difference which we present in the paper is the dissolution of the ligaments. As we know, there is no coarsening and dissolution of the topologically connected structure in ECD (i.e., the ligament size and the volume is constant during dealloying). During the LMD process, the miscible element in the ligaments will be dissolved due to the equilibrium with the decreasing concentration of the miscible element in the liquid phase, and the immiscible element in the ligaments will be dissolved due to the small but finite solubility of the immiscible element.

From the coarsening studies (Refs 30 and 31: Y. Kwon, K. Thornton, P. W. Voorhees. *EPL (Europhysics Lett.* 2009, 86, 4 46005. Y. Li, B.-N. D. Ngo, J. Markmann, J. Weissmuller. *Phys. Rev. Mater.* 2019, 3, 7 076001.), we know that the bicontinuous structure will fall apart during coarsening if the solid volume fraction is too small. For the dealloyed structure in LMD, coarsening is accelerated by the dissolution of ligaments. Therefore, if the atomic fraction of the immiscible element in the base alloy is too low, the generated topologically connected structure will be dissolved quickly and break apart. The new experimental validation is added in the main text on page 12 and Figure 5c. The measurements show that the solid volume fraction of the pure Cu melt is decreasing rapidly from the dealloying front to the edge, which indicates the fragmentation of the dealloyed structure observed in Figure 4b. In contrast, the solid volume fraction of $\text{Cu}_{70}\text{Ag}_{30}$ melt is much larger than the pure Cu melt, where most of the dealloyed structure stays connected as shown in Figure 4c.

We also note that the reviewer discussed the retention of the miscible element in the ligaments. We cited the *paper* Z. Qi, J. Weissmuller. *Acs Nano* 2013, 7, 7 5948. (Ref 25) to support our statement “ECD can produce high-genus topologically connected structures for an atomic fraction (X) of the undissolved element (e.g. Au in AgAu) as low as 5%”. While in ECD, the retention of the miscible element (Ag) is higher for 5%Au base alloy to maintain a connected structure, the story in LMD is more complicated. To address this question, we provided new analyses of the Ta concentration profiles in the ligaments on the simulations (Section 3 in the supporting information) and experiments (Page 11 and Figure 5a-b in the main text). Different from ECD, the retention of the miscible element of LMD in the dealloyed structure only depends on the melt concentrations, i.e. equilibrium concentrations on the solidus and liquidus of the

ternary phase diagram. Therefore, adding Ti into the melt will raise the Ti concentration in the solid, thereby increasing the volume fraction and maintaining the connected structures (Figure 3d). This behavior is similar to ECD. However, adding Ti into the melt will also strengthen the dissolution and dissolve the top layer of the existing structures, which is totally different from ECD. We add the measurements of the dissolution depth for various melts from the experiments in Table 1. The dissolved layer of $\text{Cu}_{80}\text{Ti}_{20}$ melt is around 190 μm , and the pure Cu melt is around 20 μm . In contrast, adding Ag into the melt is more complicated as we do not have a quaternary phase diagram. Both simulations and experiments show that the concentration of Ta in the solid does not have a significant change. Therefore, the effect of Ag addition preserving the dealloyed structure should attribute to a different source (reduced dissolution of the immiscible element Ta), rather than the retention of the miscible element Ti.

2) The discussion concerning the effects of 'Ta-leak' on the morphological evolution is rather confusing. Adding Ti would lead to a reduction of under-saturation of Ti in the Cu melt, and adding Ag could decrease the temperature of liquid metal dealloying and the degree of under-saturation; since both would alter the dealloying rate, it is not clear to the reviewer the exact effects of them on the morphology/topology of the dealloyed structure.

We understand the reviewer's concern. In ECD, dealloying kinetics controls the retention of the miscible element. However, in LMD, the retention of the miscible element is mainly controlled by the local chemical equilibrium (phase diagram). In addition, as the dealloying front velocity decreases in time, the effect of reducing the dealloying rate is revealed by the morphological evolution during dealloying. We observed that the reduced dealloying kinetics destabilizes the eutectic-like coupled growth.

Both adding Ti and Ag into the melt will reduce the dealloying kinetics. However, the different structures obtained from the Cu-Ti melt and Cu-Ag melt indicate that Ti and Ag have additional different effects on the morphology and topology of the dealloyed structures. As we discussed in the paper, adding Ti raises the Ti concentration in the liquid. The dealloyed structure has a larger volume fraction with more Ti retention in the structure but can be dissolved from the top (Table 1) due to the enhanced Ta leak effect. Adding Ag into the melt reduces the Ta leak. As shown in Figure 5c, for the similar Ta concentration in the solid, the dealloyed structure in the Cu-Ag melt has a larger volume fraction than the pure Cu melt due to the reduced overall Ta leak effect in the whole dealloying region. The

dealloyed structure from the Cu-Ag melt also benefits from Ta redeposition to remain connected at the edge of the dealloyed region.

To support our discussion of the Ta leak, we carry out a simple prediction of the volume fraction on page 12 in the main text. If we assume that there is no Ta leak (the solubility of Ta in the liquid is 0%) and Ta transport along the dealloying direction, which is more realistic for ECD, all Ta dissolved from the base alloy will be redeposited to the solid ligaments. At any dealloying depth, we have the mass conservation $c_{Ta}^s(d)S_s(d) = c_{Ta}^0(d)S_t(d)$, where $c_{Ta}^s(d)$ and c_{Ta}^0 are the Ta concentration in the ligaments and base alloy, respectively, and $S_s(d)$ and S_t are the cross-section area of the ligaments and the whole dealloying region, respectively. Therefore, the predicted volume fraction is

$\rho(d) = S_s(d)/S_t = c_{Ta}^0/c_{Ta}^s(d)$. The comparison is presented in Figure 5c in the main text. The volume fractions from the predictions are much higher than from the measurements for both the pure Cu melt and Cu₇₀Ag₃₀ melt, which directly shows the significant effect of the Ta leak.

If one of the main goals is to understand the attribution of ‘Ta leakage’ to the morphological evolution during dealloying, perhaps it would be better to simply saturate the Cu melt with Ta? This would prevent the Ta leakage and avoid affecting the rate of liquid metal dealloying of Ti, thereby more clearly test the parameter.

We agree with the reviewer’s comment that the Cu melt with saturated Ta can reduce the Ta leak. However, from the ternary phase diagram of the Ta-Ti-Cu system (Figure S1), we find that the solubility of Ta increases when the concentration of Ti increases in the Cu-Ti melt. During the LMD process, the Ti concentration near the dealloying front is much higher than the initial concentration. Therefore, we cannot prevent Ta leak due to the dissolution of Ti during the dealloying process.

2. On the modeling of phase diagram and phase field simulations:

- 1) The information provided by the phase diagrams is rather limited and incomplete in explaining the effects of Ti and Ag additions to the Cu melt on the morphology/topology.

We understand the concern of the reviewer about the simplification of the phase diagram. Regarding the practical phase diagram, the phase diagram we used is simplified to only focus on the key features and avoid complex intermetallic

phases. As we discussed in the paper, the dealloying kinetics is controlled by the dissolution of Ti, which is mainly determined by the liquidus of the Ti-Cu phase diagram. In addition, Ta leak plays an important role in the dissolution and fracture of the ligaments, which is mainly determined by the solubility of Ta in the melt. The simplified phase diagram captures these features and reproduces the solubilities of Ta and Ti in the phase-field simulations.

For the Ag addition, we agree that the assumption is not exactly correct. From the concentration profile of the experiments, we find that the ratio of the concentration of Ag and Cu is not constant. However, it's difficult to simulate a quaternary system with the Cu-Ag-Ti-Ta quaternary phase diagram. We should also note that there is no clear experimental measurement of the solubility of Ta in the Ag melt, which should be extremely small. With the pseudo-quaternary phase diagram, the results from our phase-field simulations reproduce similar dealloyed structures to the experiments.

2) The calculation method for the phase diagram appears to be incorrect? First, as shown in equation 10, the concentration terms are missing in the calculation of the grand potential.

Yes, Eq. 10 is

$$f^s - c_1^s \mu_1^s - c_2^s \mu_2^s = f^l - c_1^l \mu_1^l - c_1^l \mu_2^l$$

We have corrected it in the text.

In addition, for equations 8 & 9, the ternary phase diagrams should be calculated by equating the chemical potential of all three components, not just two?

The grand potential at the constant temperature and pressure is,

$$G = n_1 \mu_1 + n_2 \mu_2 + n_3 \mu_3$$

where n_i and $\mu_i = \partial G / \partial n_i$ are the number of moles and chemical potential of component i , respectively. If we divide the total grand potential G by the total number of moles n , we have the molar grand potential G^m ,

$$G^m = c_1 \mu_1 + c_2 \mu_2 + c_3 \mu_3$$

where $c_i = n_i / n$ is the concentration (atomic mole fraction) of component i and satisfies the constraint $c_1 + c_2 + c_3 = 1$. With this constraint, there are only two free parameters. We have the partial derivatives of G^m with respect to c_i :

$$\frac{\partial G^m}{\partial c_1} = \mu_1 - \mu_3$$

$$\frac{\partial G^m}{\partial c_2} = \mu_2 - \mu_3$$

From the above equations, the chemical potentials are,

$$\mu_3 = G^m - c_1 \frac{\partial G^m}{\partial c_1} - c_2 \frac{\partial G^m}{\partial c_2}$$

$$\mu_1 = \mu_3 + \frac{\partial G^m}{\partial c_1}$$

$$\mu_2 = \mu_3 + \frac{\partial G^m}{\partial c_2}$$

The phase equilibrium condition between the solid and liquid phases are equating the chemical potential of all three components,

$$\mu_i^s = \mu_i^l \quad i = 1,2,3$$

Substitute the chemical potentials,

$$\begin{aligned} \frac{\partial G^{m,s}}{\partial c_1^s} &= \frac{\partial G^{m,l}}{\partial c_1^l} \\ \frac{\partial G^{m,s}}{\partial c_2^s} &= \frac{\partial G^{m,l}}{\partial c_2^l} \\ G^{m,s} - c_1^s \frac{\partial G^{m,s}}{\partial c_1^s} - c_2^s \frac{\partial G^{m,s}}{\partial c_2^s} &= G^{m,l} - c_1^l \frac{\partial G^{m,l}}{\partial c_1^l} - c_2^l \frac{\partial G^{m,l}}{\partial c_2^l} \end{aligned}$$

In the phase-field model, the chemical potential is defined as $\tilde{\mu}_i = \partial G^m / \partial c_i$, $i = 1,2$. Therefore the equilibrium condition for the phase-field model is,

$$\begin{aligned} \tilde{\mu}_1^s &= \tilde{\mu}_1^l \\ \tilde{\mu}_2^s &= \tilde{\mu}_2^l \\ G^{m,s} - c_1^s \tilde{\mu}_1 - c_2^s \tilde{\mu}_2 &= G^{m,l} - c_1^l \tilde{\mu}_1 - c_2^l \tilde{\mu}_2 \end{aligned}$$

3) In related to the phase field simulation method: As the authors noted in their phase diagram modeling, there is one independent variable left. How were they able to draw the tie lines as shown in the phase diagrams? While not much details are given regarding the phase-field simulation, it is unclear how the authors handle this also in the phase field simulation.

In the ternary system at constant temperature and pressure, there are 3 equations (as shown above) and 4 variables (c_1^s , c_2^s , c_1^l and c_2^l). To draw the ternary phase diagram, we first select a concentration (for example, c_2^s), and

then calculate the rest concentrations (c_1^s , c_1^l and c_2^l) from the above equilibrium condition. In the ternary phase diagrams from the supplementary material, any thin black line connecting solidus and liquidus indicates a set of equilibrium concentrations. Different from the binary alloy solid-liquid interface, the concentration at the solid-liquid interface in the ternary alloy system is not constant but depends on the kinetics.

4) The relationship between the diffusivity and mobility given right above the equation 7 appears to be incorrect as the interdiffusion coefficient is related to the mobility. The authors can refer to: Andersson, Jan-Olof et al, Models for numerical treatment of multicomponent diffusion in simple phases, Applied Physics (1992). As shown in eq. 43 in the reference, interdiffusion coefficients would depend on the derivative of the chemical potential with respect to concentration. While this may not affect the equations solved, it does affect how the mobilities are chosen and the interpretation of the results; the thermodynamic terms can also lead to significant off-diagonal terms in the diffusivity matrix with interesting implications for the systems discussed in the paper.

The diffusivity D_{kj} defined in Eq. 43 in the reference is not the same as the diffusion coefficients defined right above the Eq. 7 in the supporting information. If we go back to the Eq. 1 and Eq. 42 in the reference, we will get,

$$\frac{\partial c_k}{\partial t} = \nabla \cdot \sum_{j=1}^n D_{kj} \nabla c_j$$

From our phase field model, we can adapt the Eq. 5 in the supporting information to,

$$\frac{\partial c_k}{\partial t} = \nabla \cdot \sum_i M_{ki} \nabla \mu_i = \nabla \cdot \sum_i M_{ki} \sum_j \frac{\partial \mu_i}{\partial c_j} \nabla c_j$$

where the second “equal” uses the chain rule. Comparing these two equations, we can obtain that the diffusivity D_{kj} in the reference is equivalent to

$\sum_i M_{ki} \frac{\partial \mu_i}{\partial c_j}$ in our phase field model, which is similar to the definition of D_{kj} in

the Eq. 43: $D_{kj} = \sum L_{ki} \frac{\partial \Phi_i}{\partial c_j}$. In the reference, the driving force Φ_i is defined in

Eq. 26: $\Phi_k = \mu_k - (V_k/V_n) \mu_n$, as there are only n-1 independent variables. In our phase field model, the chemical potential μ_i is defined as $\mu_i = \partial G^m / \partial c_i$ for

$i = 1, 2$ where $c_3 = 1 - c_1 - c_2$. If we use the same definition of μ as the reference, $\mu_{i,r} = \frac{\partial G^m}{\partial c_i}$ for $i = 1, 2, 3$, it is easy to show the relation $\mu_i = \mu_{i,r} - \mu_{3,r}$ for $i = 1, 2$.

The similar treatment can be found in other phase field models, such as the model to simulate the solidification of a multicomponent multiphase alloy system (Nestler, B., Garcke, H. & Stinner, B. *Multicomponent alloy solidification: phase-field modeling and simulations. Phys. Rev. E* 71, 041609 (2005).).

3. The comparison between the experimental and computational results are rather incomplete with some inconsistency.

1) First, experimental results did not show different alloy compositions and thus it is difficult to compare them with the computational results; the validation thus falls short.

The advantage of the computational study is that we can adjust much more parameters and run more simulations than experiments. From phase-field simulations, we found that the results vary progressively for the same added element but different alloy compositions varying from pure Cu melt to 30% Ti or 30% Ag melt. If we know the behavior of the 30%Ag melt, we can predict the result for the other compositions between the pure Cu melt and 30%Ag melt. With the guideline from phase-field simulations, we performed the experiments with the pure Cu melt and 30%Ag melt. The quantitative investigation is shown in Table 1 and Figure 5.

2) Secondly, there is a lack of quantitative analysis on the experimental data. E.g. the authors stated that the additional elements in the Cu melt results in a 'smaller and spatially more uniform ligament size', is not clear to the reviewer. Such claim can hardly be observed from the SEM micrographs presented in the manuscript.

We thank the reviewer for those suggestions. In the revised draft, we added a table (Table 1) to show the experimental measurements of the dealloying depth and some relative geometric parameters. In addition to the overview of the dealloyed structures, we discussed the effect of added elements (Ag or Ti) on the dealloying kinetics. Furthermore, we added a figure (Figure 5) to show the quantitative analysis of the experimental results.

Figure 5a-b are the concentration profiles of two experiments (pure Cu melt and Cu₇₀Ag₃₀ melt).

Figure 5c presents a comparison of the volume fraction of two experimental conditions.

Figure 5d shows the ligament size distribution with the dealloying depth. The detailed discussion of the results is in the subsections “Experimental characterization of dealloyed samples and theoretical interpretation” and “Coarsening”. There is also a section in the supplementary material (section S5). In these sections, we proposed the theoretical analysis on the non-conserved volume fraction with the dissolution of Ti and the effect of Ta leak and redeposition. We also provide a qualitative explanation of the complicated coarsening process.

3) Elemental distribution shall be analyzed (e.g. by EDS) and compared with the simulation results, especially on the retention of the Ta per discussion in the earlier comments.

Yes, we agree with the reviewer’s comments. We provide a more comprehensive analysis of the experimental results including the concentration profiles of the pure Cu melt and 30%Ag melt (Fig. 5). This information helps us to study the non-conserved volume fraction, coarsening, and fragmentation of ligaments during LMD. We also added the concentration distribution of ligaments from the phase-field simulations in Fig. S5 with the discussion in section 3 in the supplementary material.

4. On the literature associated with the dealloying field:

1) First of all, the abstract stated that ‘the past few years have witnessed widespread use of liquid metal dealloying’ may be an overstatement. While being an important, emerging dealloying method, searching on ‘liquid metal dealloying’ in Web of Science as topic yielded ~ 20 papers. Out of them about half is from Erlebacher and co-authors and the other half is from Kato and co-authors, with only few from other groups. There is much to be developed in the field and stating this method as ‘widespread’ would be misleading.

We thank the reviewer for this comment and we adapted the text as “The past few years have witnessed rapid development of liquid metal dealloying”.

2) Secondly, the manuscript misleadingly stated that ‘the recent advent of liquid metal dealloying [13]...’; while it may be fair to state that liquid metal dealloying method has only been further explored recently to fabricate nanoporous materials as cited ref. 13 in the manuscript, it is important to note that this concept was studied in a pioneer work by Harrison and Wagner back in 1959. As such, LMD method would not be a ‘recent

advent' as stated. Appropriately stating the original credit of this work would be important.

Ref. Harrison, J. D.; Wagner, C. The Attack of Solid Alloys by Liquid Metals and Salt Melts. *Acta Metall.* 1959, 7, 722–735.

We completely agree with the reviewer's comment. We used the word "recent rediscovery" to replace the "recent advent" and add the citation of Wagner and Harrison's work (Ref 13).

3) The authors stated that electrochemical dealloying is 'limited to relatively few alloys (e.g. Ag-Au or Ni-Pt)' does not seem to correctly reflect the work in the field. E.g. in the article published by the co-authors of this manuscript in *MRS Bull* in 2018 (Ref. 17), figure 1 shows that electrochemical dealloying method has been applied to more systems than the liquid metal dealloying, certainly more than the few alloys cited in this work. The authors are encouraged to present a more accurate account of the status of the field.

We completely agree with the reviewer's comment that there are various alloy systems which can be investigated by the electrochemical dealloying (ECD) method. We altered the text as "this technique limits the dealloyable alloy system (e.g. Ag-Au or Ni-Pt) to contain a relatively noble element (Au, Pt) and has a large enough difference in reduction potential to enable porosity formation." In this statement, instead of direct comparing the number of dealloyable elements between ECD and LMD, we want to demonstrate the advantage of LMD, which extends the dealloyable elements to a wider range in the period table that ECD can not process.

4) Another minor issue is that ref. 10 is not correct. It should be by McCue, Ian; Ryan, Stephen; Hemker, Kevin; et al. *Adv Eng. Mat* published in 2016, not 2015; the full citation is also missing.

We thank the reviewer for this correction. The new reference is: *I. McCue, S. Ryan, K. Hemker, X. Xu, N. Li, M. Chen, J. Erlebacher. Adv. Eng. Mater. 2016, 18, 1 46. (Ref 10)*

Reviewer #2 (Remarks to the Author):

The manuscript "Topological Control of Liquid-Metal-Dealloyed Structures" describes an interesting extension of a phase field simulation method to describe and analyse the process of liquid metal dealloying. The new aspect is the addition of a second element to the liquid metal (here by adding Ag or Ti to the Cu liquid). The authors can show in

their simulation that this alloying of the corrosion agent significantly changes the diffusion kinetics at the liquid-solid interface and the solubility of the solid phase elements (Ta and Ti) in the melt what results in different dealloying front velocities and structure connectivities. Additionally, SEM pictures of similarly dealloyed structures are shown which corroborate their main findings about the influence of the alloying of the melt.

The paper is kept quite short and compact focussing on the main finding about the influence of the composition of the melt on the process. In preceding publications of the submitting authors, especially [14] and [17], the simulation technique was developed and the structure formation discussed. Under these circumstances this extreme focus might be justified and the manuscript considered for publication. But in some instances the writing might be a bit too compact making it difficult for the reader to follow. Especially the section "Results" started with detailed compositions without explaining to the reader what exactly is done. Are they writing about a simulation or a lab experiment? What is the strategy? It becomes clear, step-by-step, in the successive paragraphs but the transition should be written in a straightforward and instantly clear way.

We thank the reviewer for this comment. We changed the text order by moving out the detailed implementation of the concentration variation. We also added a sentence to conduct the result section: "In this section, we first present the results of the investigation by phase-field simulations of the effects of adding Ti or Ag to Cu melts, which yields dramatically different morphologies." We also add subheadings to clarify the structure of the result section.

On page 4: "suggesting that diffusion-coupled growth is a ubiquitous mechanism for ECD and SMD, but not ECD that typically produces porous structures with no preferred alignment."

...ECD and SMD, but not ECD... The first ECD probably means LMD.

Yes, this is a typo. We corrected it in the text.

"the immiscible element has a vanishingly small solubility in the electrolyte and hence can only diffuse along the alloy-electrolyte interface"

I agree with the last statement. But talking about solubility of the immiscible, i.e. remaining, element during electrochemical corrosion can be misleading. In most cases, the solubility can be controlled by the applied potential. Maybe the fact that if a part of the immiscible material goes into solution, it's not going to re-deposit on the surface (unless the potential is changed) is the more decisive point.

This is a more subtle question. As the finite solubility of the immiscible element is a unique effect in LMD, we do not have a perfect analogy to ECD. There are two contributions of the finite solubility of the immiscible element during LMD. First, the dissolution of the immiscible element, which is equivalent to the

statement “If a part of the immiscible material goes into solution, it's not going to re-deposit on the surface (unless the potential is changed)”, reduces the solid volume fraction at the dealloying front. Second, the experimental result indicates the redeposition of Ta at the edge of the dealloyed structure, causing the increase of the volume fraction and preserving the existence of ligaments. For ECD with vanishingly small solubility, both effects are reduced. For the case “If a part of the immiscible material goes into solution, it's not going to re-deposit on the surface (unless the potential is changed)”, only the first effect exists. For LMD, we can observe two effects.

On page 7: "the spacing λ of solid ligaments at the dealloying front obeys approximately the $\lambda^2 V = C$ law for stable growth [14]"
This finding is only cited in reference [14]. The authors should cite the original studies.

We thank the reviewer for this suggestion and we added a new reference paper about the eutectic growth: *K. Jackson, J. Hunt. In Dynamics of Curved Fronts, 363–376. Elsevier, 1988. (Ref 29)*

On page 8: "corresponds to a reduction of Peclet number P by a factor of 3.3 compared to about 5.4 in the phase-field simulations."
The description of the character of the transport phenomena (flow and diffusion) by means of the Peclet number at that point is completely unmotivated. The authors should explain the reader what exactly the Peclet number describes in that scenario and what the observed change means.

We agree with the reviewer’s comment. The definition of Peclet number for the 1D dealloying kinetics can be found in this paper: *Ian McCue, Bernard Gaskey, Pierre-Antoine Geslin, Alain Karma, and Jonah Erlebacher. Kinetics and morphological evolution of liquid metal dealloying. Acta Materialia, 115:10–23, 2016 (Ref 16)*. We adapted the calculation to our 2D and 3D dealloying simulations. The Peclet number p rates the dealloying kinetics in the form:

$$x_i(t) = -\sqrt{4D_l p t}$$

where $x_i(t)$ is the dealloying depth and D_l is the liquid diffusivity. We have added the full calculation in the supplementary material, section 4.

On page 10: "purified Ar atmosphere": Just a technical note: Purified by which means?

We thank the reviewer for pointing out this misleading description. Since we did not perform any additional Ar purification, we changed the phrase to “high-purity (99.999%) Ar atmosphere”.

Reviewer #3 (Remarks to the Author):

In this paper, influence of adding element to a metallic melt used for the dealloying medium on porous structure evolution during the liquid metal dealloying (LMD) is discussed both computationally and experimentally to find methods to reduce the parting limit concentration (X) of the remaining element for constructing the bi-continuous structure and to suppress the ligament growth.

The author group has reported that dipping a Ti-Ta alloy precursor in a Cu melt results in the selective dissolution of Ti element, thus in the self-organizing bi-continuous porous Ta in a Cu melt. In this dealloying system, Ti and Ag were selected and added in the Cu melt separately. The phase field model simulation demonstrated that (1) the Ti addition suppressed Ti dissolution but enhanced the Ta leakage to the melt, resulting in decreasing the dealloying rate, increasing the parting limit X , and suppressing the ligament alignment by the pinch off. On the other hand, (2) the Ag addition in the melt suppressed the Ta leakage, resulting in dramatically decreasing X , and enhancing the ligament alignment by suppressing the pinch off. The different effects of these additions are successfully confirmed in the experimental counterparts.

Although the paper is well organized with the data by the reliable computational and experimental protocols, and the finding in this study actually suggests methods to control the porous structure in LMD, the method itself is not special and within the range of expectations, i.e. varying dealloying temperature or time, and adding element(s) to precursor, etc., so, does not have enough impact for publication in Nature Communications.

We thank the reviewer for those comments. However, we respectfully disagree with the reviewer’s assessment that our paper does not have enough impact for publication in Nature Communications. This paper provides insight into the fundamental mechanism of controlling the topologically connected structures during liquid metal dealloying. We provide a method by varying the melt to alter the dealloying kinetics and topology of the dealloyed structures. Moreover, we discover the leak of the immiscible element (Ta) in LMD, which is a fundamental difference between ECD and LMD. We explore the effect of Ta leak on the evolution of the topology and morphology of the dealloyed structures computationally and experimentally. Therefore, our paper provided insight into the mechanism of LMD and an effective guideline for material scientists to optimize their LMD techniques to obtain the desired bicontinuous structures.

P4, line8: “ECD and SMD, but.....” should be “LMD and SMD, but.....”

Yes, this is a typo. We corrected it in the text.

P8, line19: It is mentioned that the Ta (a typo, should be Ag) addition to the Cu melt reducing the ligament growth rate by the bulk diffusion of Ta in the Cu liquid. It is better to mention why the bulk diffusion of Ta in the Cu melt influences the ligament growth.

We thank the reviewer for this suggestion. In this revised draft, we present a more comprehensive investigation of the dealloyed structures from experiments in the subsections “Experimental characterization of dealloyed samples and theoretical interpretation” and “Coarsening”. In the supporting information, we added a section discussing the coarsening of ligaments (Section 5). The discussion of the ligament growth is developed to explore the effect of Ta leak on coarsening of the dealloyed structures.

As we discussed in the draft, different from ECD, the dealloyed structure coarsens during liquid metal dealloying. Fundamentally, for the bulk-diffusion driven coarsening, bulk diffusion of Ta dominates the coarsening kinetics, as the bulk diffusion rate of Ta is much lower than Ti. However, the power-law check (Fig. S8) shows that the coarsening process in LMD does not follow the standard scaling law for capillary-driven coarsening, neither surface-diffusion driven nor bulk-diffusion driven. Our research reveals that two more effects influence the coarsening. First, the ligaments are dissolved during coarsening as we found that the solid volume fraction during LMD is decreasing. Second, Ta leak and redeposition, which are not capillary-driven, preserve the dealloyed structure at the edge. We provided a qualitative analysis in this draft but a quantitative model of coarsening in LMD is beyond our current work.

Reviewers' comments:

Reviewer #2 (Remarks to the Author):

I think the manuscript has greatly improved. The new structure of the section "Results and Discussion" and the added details and deeper analysis of the consequences of the Ta leak are well discussed. I hope nature communications allows the necessary space for this new amount of text.

All my points have been addressed. I recommend a publication in nature communication.

Reviewer #3 (Remarks to the Author):

The simulation and experimental results that suppressing Ta dissolution into Cu liquid by Ag addition in the liquid preliminary can construct the porous structure even for a low concentration (X) of the ligament element Ta in the precursor is interesting and meaningful. However, this finding is valid for specific case that ligament element leakage happens actively during LMD, such in the present Ta-Ti-Cu system, thus not general.

The adding Ag may also influence dissolution of sacrificing Ti element from the reaction front. Therefore, the final concentration of Ti in the ligament should be discussed in the text. The Ta concentration in the ligament is discussed in Fig. 2, but that of Ti is not clear in this figure. If the remaining Ti concentration in the ligament after LMD did not reach ≈ 0 at.% due to the Ag addition, it is not surprising to the reviewer that the porous structure is obtained from the precursor with such low Ta concentration ($X = 0.15$) < the reported parting limit $X = 0.3$.

This paper demonstrates interesting simulation and experimental results. However, considering the above, the reviewer still does not think that this paper has enough scientific impact for publication in Nature Comm.

Reviewer #4 (Remarks to the Author):

The major motivation for this work seems to be highlighting and describing what the authors perceive to be some fundamental differences between electrochemical dealloying (ECD) and liquid metal dealloying (LMD) for the production of nanoporous structures. In this respect the authors make two primary claims and several more minor claims:

Claim 1: First, dealloying kinetics in ECD is interface controlled with a constant dealloying front velocity V that depends of the applied voltage.

This claim is not correct: Chronoamperometry experiments in ECD demonstrate that dealloying does not occur at a constant rate as there is a clear, albeit slow, decrease in current density with time. If, however, one examines this decay over relatively short periods of time, as was the case in several x-ray publications from the Northwestern group (see for example, Acta Mat., 61, 1118, 2013 and Acta Mat., 61, 5561, 2013) the dealloying rate and the dealloying front velocity appears to be constant. Under typical conditions of ECD, the dealloying rate is under activation control or in the language the present authors use, interface controlled. This is because liquid-

phase mass transport is relatively fast in comparison to the slower activation-controlled rate employed in many dealloying protocols. Consider as the present authors do, dealloying of a single-phase AXB_{1-X} alloy in which the B component is selectively dissolved. At large enough driving force (voltage) the rate of B dissolution becomes faster than the rate at which B can be transported from the solid-liquid interface. As the concentration of B builds up in the interfacial region, sooner or later the solubility of B in the electrolyte is exceeded. In fact, the electrolyte eventually becomes supersaturated in B and a salt-film precipitation event occurs. At this stage, the rate limiting step for B dissolution is clearly in the electrolyte. Another way of demonstrating this is simply to do an experiment by saturating the electrolyte in the B-component. For this initial condition, very quickly the electrolyte will become supersaturated and the rate limiting step will transition to liquid-phase mass transport. A third way of thinking about the decay in current during ECD is the following. As the dealloying front penetrates into the solid, the path length in the porous liquid phase increases. Since the liquid has a finite ionic conductivity, there is an IR drop occurring with increasing path length. Regardless of the magnitude of the current, I, the voltage at the dealloying front that provides the driving force for dealloying must decay with time. Interestingly, a full analysis of this shows that the current decays as 1/Sqrt (time), similar to the velocity decay in LMD!

Claim 2: Second, in ECD, the immiscible element has a vanishingly small solubility in the electrolyte and hence can only diffuse along the alloy-electrolyte interface.

Unfortunately, the authors base their arguments connected to ECD on alloy systems for which the A component is virtually insoluble; noble-metal containing alloys such as AgAu, CuAu, NiPt etc. and only aqueous electrolytes. However, nanoporous structures are known to evolve during dealloying of systems such as CuMn, NiMn, NiAl, CuAl, CuZn and a host of other alloys. The point is that for these alloy systems both components are soluble in the electrolyte and by controlling the voltage one can minimize (but often not entirely eliminate) the dissolution of the more noble A-component. Additionally, as described in the manuscript by Harrison and Wagner, ECD in molten salts provide another route for nanoporous metal production that significantly widens the window of alloys that can be used in ECD (e.g. FeNi) in comparison to what is possible in aqueous electrolytes. The point is that under certain alloy-electrolyte combinations the differences between LMD and ECD discussed in the manuscript disappear.

In my view, under typical conditions, the major difference between LMD and ECD dealloying is related to the orders of magnitude difference in dealloying rates. In ECD, one can control the dealloying rate by adjusting the temperature, voltage and electrolyte composition. In LMD, one can control the liquid composition and the temperature; the most sensitive knob, the voltage is lost. It is therefore expected that if the rate of LMD is reduced, an alloy containing only 5%A can show a bicontinuous structure in analogy to what is observed in ECD of a Ag 5 at% Au alloy at low enough dealloying rates.

Another major problem I have with this manuscript is the comment (pg. 15 lines 332-333), "This is in contrast to ECD, where coarsening is essentially absent during dealloying but can be induced by annealing at higher temperature after dealloying."

Why do these authors believe that during ECD coarsening is not occurring behind the dealloying front? Consider dealloying of a Ag₂₅%Au alloy in perchloric acid. At the dealloying front the

nanoporous gold ligaments forming are of order a 2-3 nanometers. For such gold structures, there is a well-known exponential decrease of the melting point as a result of the large surface to volume ratio (see Buffat and Borel, "Size effect on the melting temperature of gold particles", Phys. Rev. A, 13, 2287, 1976 and many more recent publications on this topic). For 3 nm diameter gold particles the melting point is about 525 K so that at 300 K, the homologous temperature is 0.57. Accordingly, for these small length scales, coarsening occurs by a combination of surface and bulk diffusion! By the time the structure coarsens to 5 or 6 nm the melting point is close to 1000 K so that at this size coarsening will only be occurring by surface (interface) diffusion. Obviously, all of this must be occurring behind the dealloying front during ECD.

As the authors describe, addition of Ti to the Cu melt will slow down the rate of LMD. Frankly, it's unclear why they didn't virtually saturate the Cu melt with Ti in order to slow the dealloying rate so that they could observe connected dealloyed morphologies at significantly reduced dealloying depths. That said, the effect of Ti in the Cu melt was examined by PF simulations, but remarkably, no experimental results were actually shown for this. While Table 1 indicates Cu_{0.80}Ti_{0.20} were employed in experiments, Figures 4 and 5 only show results for Ag added to the Cu melt. This is incomprehensible to me since they claim that the experiments verify their simulation results. Importantly, experimental results for adding Ti to the Cu melt should be simpler to understand and explain compared to adding Ag to the Cu melt.

Finally, a comment on the experimental results. While the manuscript discusses simulated dealloying morphologies in detail, the corresponding experimental results (Fig. 4 b and c) are at best insufficient. The SEM showing the 10-micron bar for images 4b and 4c is at too low a resolution for comparison with the simulation results. Why was this not examined at much higher magnifications? Also, Figure 5 shows error bars for experimental results. Two questions in this regard: 1) What was the reproducibility of these results with respect to the number of samples examined and 2) what is the basis of the error bars?

There are some more trivial issues connected to the manuscript, but given my concerns regarding the major claims of the manuscript those issues are inconsequential. Also, in comparing the contents of this manuscript to their 2015 Nat. Comm. paper and the 2016 Acta Mat. paper, there is not much that is new and publishable that is supported by experimental results.

Responses (black text) to reviewers's comments (blue text) for "Topological Control of Liquid-Metal-Dealloyed Structures", by Lai et al.

Reviewers' comments:

Reviewer #2 (Remarks to the Author):

I think the manuscript has greatly improved. The new structure of the section "Results and Discussion" and the added details and deeper analysis of the consequences of the Ta leak are well discussed. I hope nature communications allows the necessary space for this new amount of text.

All my points have been addressed. I recommend a publication in nature communication.

We thank the reviewer for recommending our manuscript for publication as is.

Reviewer #3 (Remarks to the Author):

The simulation and experimental results that suppressing Ta dissolution into Cu liquid by Ag addition in the liquid preliminary can construct the porous structure even for a low concentration (X) of the ligament element Ta in the precursor is interesting and meaningful. However, this finding is valid for specific case that ligament element leakage happens actively during LMD, such in the present Ta-Ti-Cu system, thus not general.

We understand that the reviewer may have reached this conclusion from the current text. However, we have added a paragraph to the discussion section to emphasize that the major novel insights of our paper are expected to be general and apply to other related methods such as solid-state dealloying (SSD). Prior to our work, the effect of the leak of the immiscible element on LMD structures had been completely ignored. This is largely because this effect is negligible in ECD, and so far LMD was naively believed to be similar to ECD. This is generally not the case. Our paper reveals the crucial role of this leak by modifying the melt composition to control the rate of leakage. Granted, the experiments were carried out for a specific base alloy system (TaTi) and by adding a specific element Ag to a Cu melt to reduce the leakage rate. However, the insights derived from those experiments are general. Those insights are embodied in Equation (2) that predicts quantitatively for any alloy system how the leak of the immiscible element reduces the solid volume fraction at the dealloying front, thereby explaining why structures dealloyed in different melts can exhibit very

different topologies. A crucial difference between ECD and LMD is that, in LMD, the solubility of the immiscible element in the liquid is significantly enhanced by the high concentration of the miscible element on the liquid side of the interface (c_{Ti}^l here), which in turn increases the concentration of the immiscible element (c_{Ta}^l here) on the liquid side of the interface and reducing the solid volume fraction as predicted by Equation (2). This enhancement is due to the fact that the solid-liquid interface during LMD is in local thermodynamic equilibrium, such that a high c_{Ti}^l contributes to raising c_{Ta}^l . In contrast, during ECD, electrochemical removal of Ag from a AgAu alloy is a non-equilibrium reaction that does not increase the solubility of Au in the electrolyte. Beyond LMD, we also expect our results to apply to SSD where the solid-solid interface is expected to remain in local thermodynamic equilibrium during dealloying. This expectation is supported by the fact that the solid volume fraction is observed to vary across the dealloyed layer of SSD structures. For example, the SEM Figure 1 in Ref. [1] shows that the dealloyed structure is dense at the dealloying front but much more sparse at the edge of the dealloyed layer, thereby implying that dissolution of the solid ligaments associated with leakage of the immiscible element is occurring during dealloying.

1: Wada, Takeshi, Kunio Yubuta, and Hidemi Kato. "Evolution of a bicontinuous nanostructure via a solid-state interfacial dealloying reaction." *Scripta Materialia* 118 (2016): 33-36.

We have added new text from line 305 to 326 of the revised manuscript to explain the generality of our results.

The adding Ag may also influence dissolution of sacrificing Ti element from the reaction front. Therefore, the final concentration of Ti in the ligament should be discussed in the text. The Ta concentration in the ligament is discussed in Fig. 2, but that of Ti is not clear in this figure.

Since Ta and Cu have a high mixing enthalpy, the concentration of Cu in the solid ligament is relatively small (few percent at most, see the attached figures). Therefore, to a good approximation, the concentration of Ti in the solid ligament is given by $c_{\text{Ti}}^s = 1 - c_{\text{Ta}}^s$. When discussing the volume fraction, the concentration that we are really concerned with is the retention of all the elements except the immiscible element, which by mass conservation is exactly $c_r = 1 - c_{\text{Ta}}^s$. Since we already show the concentration of Ta in the solid, c_{Ta}^s , it is not necessary to also show c_r^s in the same figure.

If the remaining Ti concentration in the ligament after LMD did not reach $\approx 0\text{at.}\%$ due to the Ag addition, it is not surprising to the reviewer that the porous structure is obtained from the precursor with such low Ta concentration ($X=0.15$) < the reported parting limit $X=0.3$.

While we agree with the reviewer that it is not, in and of itself, surprising that the solid volume fraction can remain above the threshold for fragmentation for $X=0.15$ due to retention of the miscible element in the dealloyed structure (and also some one of the melt elements, here Cu), what is surprising, or at least non

trivial to us, is that the amount of retention can be dramatically altered by the choice of melt for reasons that are revealed and explained for the first time in our work. This altered retention then impacts the solid volume fraction and the topology of the dealloyed structure. In fairness to the author, we urge the reviewer to please read carefully the response to the reviewer's first comment above and reflect on the fundamental differences between ECD and LMD (also SSD) revealed by our work.

This paper demonstrates interesting simulation and experimental results. However, considering the above, the reviewer still does not think that this paper has enough scientific impact for publication in Nature Comm.

We hope that the reviewer now judges our paper sufficiently impactful for Nature Comm. given the generality of our results emphasized in the response above to the reviewers first comment and the new text on lines 305 to 326.

Reviewer #4 (Remarks to the Author):

The major motivation for this work seems to be highlighting and describing what the authors perceive to be some fundamental differences between electrochemical dealloying (ECD) and liquid metal dealloying (LMD) for the production of nanoporous structures. In this respect the authors make two primary claims and several more minor claims:

The reviewer seems to have only made a cursory read of the paper, the Claims the reviewer identifies as central to the paper are only ancillary observations, and the critiques of these Claims are fraught with basic factual errors.

Claim 1: First, dealloying kinetics in ECD is interface controlled with a constant dealloying front velocity V that depends of the applied voltage.

This claim is not correct: Chronoamperometry experiments in ECD demonstrate that dealloying does not occur at a constant rate as there is a clear, albeit slow, decrease in current density with time. If, however, one examines this decay over relatively short periods of time, as was the case in several x-ray publications from the Northwestern group (see for example, Acta Mat., 61, 1118, 2013 and Acta Mat., 61, 5561, 2013) the dealloying rate and the dealloying front velocity appears to be constant. Under typical conditions of ECD, the dealloying rate is under activation control or in the language the present authors use, interface

controlled. This is because liquid-phase mass transport is relatively fast in comparison to the slower activation-controlled rate employed in many dealloying protocols. Consider as the present authors do, dealloying of a single-phase AXB_{1-X} alloy in which the B component is selectively dissolved. At large enough driving force (voltage) the rate of B dissolution becomes faster than the rate at which B can be transported from the solid-liquid interface. As the concentration of B builds up in the interfacial region, sooner or later the solubility of B in the electrolyte is exceeded. In fact, the electrolyte eventually becomes supersaturated in B and a salt-film precipitation event occurs. At this stage, the rate limiting step for B dissolution is clearly in the electrolyte. Another way of demonstrating this is simply to do an experiment by saturating the electrolyte in the B-component. For this initial condition, very quickly the electrolyte will become supersaturated and the rate limiting step will transition to liquid-phase mass transport. A third way of thinking about the decay in current during ECD is the following. As the dealloying front penetrates into the solid, the path length in the porous liquid phase increases. Since the liquid has a finite ionic conductivity, there is an IR drop occurring with increasing path length. Regardless of the magnitude of the current, I, the voltage at the dealloying front that provides the driving force for dealloying must decay with time. Interestingly, a full analysis of this shows that the current decays as 1/Sqrt (time), similar to the velocity decay in LMD!

The notion of diffusion-limited dissolution in the electrochemical environment is perhaps operative at wildly long timescales and dealloying depths. This is not the regime of interest here, or in any practical situation we are aware of. The paper cited from the Northwestern group clearly shows a linear increase of the depth with time over tens of microns dealloying depth (many orders of magnitude larger than the ligament size)!

The reviewer is positing a hypothetical that is nonsense, in considering an electrochemical system where the solute builds up near the moving interface at an unpractically high voltage and slow transport rates. In practice, actual oxidation, oxygen evolution, or other electrochemical effects will occur well before the driving force could ever be that high. A simple back-of-the-envelope calculation also indicates the conclusion about mass transport rates is also nonsense: in electrolyte, a typical ion diffusion coefficient is 10^{-6} cm²/s, and a typical dealloying rate is 20 nm²/s (, Acta Mat., 61, 1118, 2013). The average diffusion distance $x \sim \sqrt{D_{l}t}$, so in one second a dissolved ion will move 1000 microns, while the interface only moves 20 nm.

We are frankly astounded that the “claim” bothers the reviewer – it is correct and well-established in the literature. Most pertinently, it also isn’t a novel claim that we make in the paper, nor a relevant one to critique. Under the conditions we are interested in, ECD exhibits interface-limited kinetics and LMD exhibits diffusion-limited kinetics, and although we set up a comparison/contrast to put our innovation in context, our paper is about the implications of diffusion-limited kinetics in LMD leading to a richer and more complex interaction between the kinetics and the topology of the growing porous phase.

Claim 2: Second, in ECD, the immiscible element has a vanishingly small solubility in the electrolyte and hence can only diffuse along the alloy-electrolyte interface.

Unfortunately, the authors base their arguments connected to ECD on alloy systems for which the A component is virtually insoluble; noble-metal containing alloys such as AgAu, CuAu, NiPt etc. and only aqueous electrolytes. However, nanoporous structures are known to evolve during dealloying of systems such as CuMn, NiMn, NiAl, CuAl, CuZn and a host of other alloys. The point is that for these alloy systems both components are soluble in the electrolyte and by controlling the voltage one can minimize (but often not entirely eliminate) the dissolution of the more noble A-component. Additionally, as described in the manuscript by Harrison and Wagner, ECD in molten salts provide another route for nanoporous metal production that significantly widens the window of alloys that can be used in ECD (e.g. FeNi) in comparison to what is possible in aqueous electrolytes. The point is that under certain alloy-electrolyte combinations the differences between LMD and ECD discussed in the manuscript disappear.

The reviewer has missed a basic and critical point in dealloying. Sure, there are systems with differential solubility of the two components in the electrolyte in ECD. There are as well in LMD or molten salt dealloying. Our contribution is to note a new effect – that the dissolving species itself (Ti here) can locally modify the solubility of the other component (Ta) leading to a transient solubility near the moving interface. No one has ever suspected this before – we identify the effect and point out the profound implications it has on the morphological evolution.

In my view, under typical conditions, the major difference between LMD and ECD dealloying is related to the orders of magnitude difference in dealloying rates. In ECD, one can control the dealloying rate by adjusting the temperature, voltage and electrolyte composition. In LMD, one can control the liquid

composition and the temperature; the most sensitive knob, the voltage is lost. It is therefore expected that if the rate of LMD is reduced, an alloy containing only 5%A can show a bicontinuous structure in analogy to what is observed in ECD of a Ag 5 at% Au alloy at low enough dealloying rates.

Another major problem I have with this manuscript is the comment (pg. 15 lines 332-333), “This is in contrast to ECD, where coarsening is essentially absent during dealloying but can be induced by annealing at higher temperature after dealloying.”

Why do these authors believe that during ECD coarsening is not occurring behind the dealloying front? Consider dealloying of a Ag25%Au alloy in perchloric acid. At the dealloying front the nanoporous gold ligaments forming are of order a 2-3 nanometers. For such gold structures, there is a well-known exponential decrease of the melting point as a result of the large surface to volume ratio (see Buffat and Borel, “Size effect on the melting temperature of gold particles”, Phys. Rev. A, 13, 2287, 1976 and many more recent publications on this topic). For 3 nm diameter gold particles the melting point is about 525 K so that at 300 K, the homologous temperature is 0.57. Accordingly, for these small length scales, coarsening occurs by a combination of surface and bulk diffusion! By the time the structure coarsens to 5 or 6 nm the melting point is close to 1000 K so that at this size coarsening will only be occurring by surface (interface) diffusion. Obviously, all of this must be occurring behind the dealloying front during ECD.

We believe the ridiculousness of this claim should discount the opinion of the reviewer. Interface diffusion at the gold/electrolyte interface of course leads to a small amount of coarsening near the interface, growing the length scale of porosity in ECD from 2-3 nm at the growing interface to 10-20 nm within a few nanometers, at which point the length scale is essentially kinetically stuck and further coarsening is absent. But it isn't very fast relative to LMD, and this is why there isn't a gradient in pore size in ECD like there is in LMD. Even after a day in acid, the pore size in acid only grows to ~50 nm (see Advanced materials 16 (21), 1897-1900). Only by increasing the temperature does one get significant coarsening.

The connection to thermodynamic melting point depressions is bizarre. Of course there is a decrease in the melting point of nanoparticles. It is why nucleation during solidification happens and has been known for over a century! But we don't have nanoparticles. We have a bicontinuous structure, and porous gold has a zero mean curvature, as we've shown by TEM tomography (Applied Physics Letters 92 (25), 251902). Systems with zero mean curvature don't have

a melting point depression – it is why microporous materials don't spontaneously melt. There is absolutely no evidence of bulk diffusion or local melting in any literature on typical dealloying, the most obvious observation supporting this being that the grain structure of the parent alloy is maintained. But let us humor the thought experiment posed by the reviewer. The works cited are all particles of gold in vacuum, for which the surface energy is well over 4 J/m^2 . In electrolyte, we've measured (and reported in Nature materials 5 (12), 946-949) the surface energy is a lot lower, 1.2 J/m^2 . This because in electrolyte under potential control, the interface is comprised of gold and adsorbed ions in the double layer, as any electrochemist will confirm. We find that a 3 nm curvature particle would have a homologous temperature of 0.51, i.e., lower than suggested. Materials at 51% of their melting point (or 57% for that matter) do not have significant vacancy concentrations. For gold, with a formation energy of 0.83 eV/vacancy, we'd expect a mol fraction of vacancies at this temperature to be $\sim 10^{-8}$. Unless the reviewer is suggesting that the formation energy of vacancies magically decreases by an order of magnitude, the concentration of vacancies is irrelevant.

Taken together, we suspect the reviewer is harkening back to theories of Pickering and Wagner from the 1960s that invoked exotic "divacancies" explain porosity evolution, theories which were relevant at the time when no other model for porosity evolution existed, but have now been thoroughly discounted in lieu of the interface diffusion-controlled models that we have developed and published a number of papers in Nature family journals.

[the reviewer may be thinking of dealloying studies by Sieradzki and co-workers in low melting systems (Li-Pb) which in fact do have a contribution from bulk vacancies, but that is a special case]

As the authors describe, addition of Ti to the Cu melt will slow down the rate of LMD. Frankly, its unclear why they didn't virtually saturate the Cu melt with Ti in order to slow the dealloying rate so that they could observe connected dealloyed morphologies at significantly reduced dealloying depths. That said, the effect of Ti in the Cu melt was examined by PF simulations, but remarkably, no experimental results were actually shown for this. While Table 1 indicates $\text{Cu}_{0.80}\text{Ti}_{0.20}$ were employed in experiments, Figures 4 and 5 only show results for Ag added to the Cu melt. This is incomprehensible to me since they claim that the experiments verify their simulation results. Importantly, experimental results for adding Ti to the Cu melt should be simpler to understand and explain compared to adding Ag to the Cu melt.

Table 1 reports the experimental results for $\text{Cu}_{80}\text{Ti}_{20}$ melts and we would be happy to add a corresponding experimental image to the supplementary

material. However, most of our paper focuses on the much more interesting case of Ag addition to the melt that reduces the Ta leak with a profound impact on the topology of dealloyed structures.

Finally, a comment on the experimental results. While the manuscript discusses simulated dealloying morphologies in detail, the corresponding experimental results (Fig. 4 b and c) are at best insufficient. The SEM showing the 10-micron bar for images 4b and 4c is at too low a resolution for comparison with the simulation results. Why was this not examined at much higher magnifications? Also, Figure 5 shows error bars for experimental results. Two questions in this regard: 1) What was the reproducibility of these results with respect to the number of samples examined and 2) what is the basis of the error bars?

The reviewer has clearly missed that, in both the 2015 Nat. Comm. and the present paper, the phase-field simulations explore orders of magnitude shorter time scales than the 10 sec duration of the experiments. Therefore, on the simulation time scale, both the dealloying depth and the ligament size are much smaller than on the SEM images and increasing the resolution of those images would not provide a basis of comparison of simulation and experimental results.

There are some more trivial issues connected to the manuscript, but given my concerns regarding the major claims of the manuscript those issues are inconsequential. Also, in comparing the contents of this manuscript to their 2015 Nat. Comm. paper and the 2016 Acta Mat. paper, there is not much that is new and publishable that is supported by experimental results.

The major claims 1 and 2 that the reviewer has chosen to criticize have to do with ECD, and not with LMD that is the main focus of our paper. We only invoke ECD to highlight important fundamental differences between the two processes in the light of our results. If the statements we make about ECD, when comparing ECD and LMD, were wrong, the reviewer's criticisms would be relevant. However, those criticisms are egregiously erroneous as detailed in our responses above. Our paper goes well beyond the 2015 Nat. Comm by demonstrating unambiguously the crucial role of the leakage of the immiscible (i.e. least soluble) element of the base alloy into the liquid melt.

Reviewer #5 (Remarks to the Author):

I read this paper, and the 2015 paper by these authors in Nature Comm. I found the topic of LMD interesting, and the approach of integrating modelling and experiment innovative. The first paper elucidates the physical mechanism of length scaling in liquid metal de-alloying, while the present manuscript demonstrate how to promote high-genus (i.e. more complex) topologies by limiting the leakage of the immiscible element into the liquid, which in turn can be done by controlling the diffusion of the immiscible element in the liquid. In that sense the first paper is introducing the effect to a broader audience and the second is showing a design path for using this process (e.g. for designing high porosity materials relevant to batteries, etc.) under somewhat general situations (even though it demonstrated for one specific set of alloys, the generality is discussed in the manuscript).

The mathematics and interpretation of graphs, scaling plots, etc are technically sound, and what is presented in the main text is at sufficiently basic level that the general reader can appreciate, with any esoteric aspects of modelling being relegated to the supplementary material.

I also read the full exchange between the authors and the reviewers. Reviewer 2 and 3 in particular both appear to appreciate the work and find it acceptable for publication. Reviewer 4 however, is against publication and brings up criticisms against two claims made in the present manuscript (one of them not really made in this manuscript). The reviewer further believes that there is not much difference between this work and the 2015 paper. With regard to this last point, Reviewer 4 is not correct. As I mentioned above, this manuscript is clearly different from the one in 2015 in that it aims to elucidate topology control, while the first paper exposes the essential physics of de-alloying, how the scaling cell spacing arises, etc. As to the main criticisms of Reviewer 4, the reviewer appears to confuse aspects related to the ECD process with those in liquid metal de-alloying to electrochemical de-alloying. Moreover, the reviewer's response appears to suggest that they feel that the present manuscript is criticizing ECD, which it is not. This manuscript is specifically about controlling topology in de-alloying through the diffusion of the immiscible element in the liquid through, a diffusion-limited process that occurs on different time and length scales than ECD. As such, the comparison (or confusion?) of the two processes doesn't seem relevant as a critique of the present manuscript main point.

I believe the manuscript is technically sound, probes scaling laws and patterning in natural process that is also of sufficient practical importance to be of interest to the broader audience of Nature Comm. I thus recommend its publication.